# HERON: Human-robot collaboration with Efficient and Resilient OptimizatioN for Long-horizon planning

## Abstract

The integration of humans into long-horizon planning introduces unique challenges that extend beyond conventional robotic task planning. Unlike robots, humans exhibit inherent uncertainty in task execution, including variable performance, unexpected interruptions, and dynamic goal changes, all of which complicate efficient collaboration. To address these challenges, we propose Human-robot collaboration with Efficient and Resilient OptimizatioN for Long-horizon planning (HERON), a novel framework that combines large language models (LLMs), physics-guided reasoning, and optimization techniques. HERON leverages LLMs in two complementary roles: (i) decomposing natural language task descriptions into structured sub-tasks with agent assignments, and (ii) generating physics-guided execution time estimates and determining sub-task assignments for both human and robot agents based on physical constraints and complementarities. These outputs are incorporated into a mixed-integer linear programming scheduler, which dynamically re-schedules based on observed human uncertainties. This integration ensures that scheduling is not only feasible with respect to physical limitations but also robust to human unpredictability while maintaining efficiency in resource and time allocation. Experiments demonstrate that HERON enables resilient and adaptive human-robot collaboration, achieving more efficient scheduling and higher task success rates compared to existing LLM-based planning frameworks. Website at https://sites.google.com/view/heron-planner.

## 1 Introduction

Long-horizon planning lies at the heart of enabling embodied agents to tackle complex tasks that extend across multiple steps, dependencies, and temporal constraints (Bacchus & Kabanza, 1998; Nau et al., 2003). Unlike short-horizon or reactive decision-making, long-horizon tasks require structured decomposition into sub-goals and careful coordination to ensure overall success (Yang et al., 2025b; Mavrogiannis et al., 2024). With the advent of large language models (LLMs), recent approaches have leveraged their reasoning and compositional strengths to perform task decomposition, agent allocation, and plan generation in natural language form (Singh et al., 2023; Yoneda et al., 2024; Kannan et al., 2024; Zhao et al., 2024; Nayak et al., 2024; Zhang et al., 2025; Liu et al., 2025b; Yang et al., 2025a; Wu & Xiao, 2025; Chen et al., 2025; Zhu et al., 2025). Despite their versatility and ability to generalize across diverse tasks, however, LLM-based planning often struggles with efficiency and reliability. For example, plans generated by LLMs are frequently verbose, contain redundant steps, or fail to optimize resource usage, especially in long-horizon settings. Moreover, while LLMs can produce coherent high-level strategies, they often lack mechanisms to incorporate domain-specific constraints, such as physical feasibility or timing, leading to inefficiencies when applied to embodied agents in realistic environments (Obata et al., 2024).

Building on this foundation, researchers have increasingly sought to extend long-horizon planning into human-robot collaboration (HRC), motivated by the unique strengths of combining human flexibility with robotic precision (Hoffman & Breazeal, 2004; Goodrich et al., 2008; Pirk et al., 2020). While such studies demonstrate the potential of shared autonomy, they often adopt a human-centric paradigm, where the human drives decision-making and the robot serves primarily as a limited assistant. In these scenarios, robots are often limited to execution-level support—fetching objects,

following instructions, or performing narrowly defined actions—without engaging in planning or adaptive coordination (Huang et al., 2024; Liu et al., 2025a; Huang et al., 2025). This imbalance constrains the robot's role and fails to realize the full promise of collaborative long-horizon planning.

More recent attempts have aimed to design frameworks for efficient HRC in long-horizon tasks, explicitly considering task allocation and scheduling between humans and robots (Chen et al., 2024; Yu et al., 2024; Izquierdo-Badiola et al., 2024; Gao et al., 2024). However, these frameworks tend to be overly specialized and domain-dependent, relying on heavily engineered assumptions that hinder generalization to diverse collaboration settings. Moreover, they rarely incorporate systematic models of human uncertainty, such as variability in task execution time, unexpected pauses, or dynamic goal changes—all of which are central to real-world HRC (Ahn et al., 2024). As a result, existing approaches remain ill-suited to environments where resilience and adaptability are required.

To address these gaps, we propose Human-robot collaboration with Efficient and Resilient OptimizatioN for Long-horizon planning (HERON), a framework designed to make scheduling in HRC both physically grounded and uncertainty-aware. First, an LLM-based planner translates natural language goals into a set of sub-tasks with explicit temporal dependencies. Second, a physics-guided LLM estimates execution times by reasoning over kinematics, actuation limits, and environmental factors, while simultaneously assigning tasks to human or robot agents based on feasibility and complementarity. Third, a Mixed-Integer Linear Programming (MILP) optimizer (Perron & Furnon, 2019) generates efficient schedules that respect dependencies and constraints, ensuring effective resource and time allocation. In addition to these three core modules, HERON includes a verifying stage that monitors execution, detects human uncertainty events, and triggers re-planning when necessary. Together, these components allow HERON to move beyond one-shot planning: the system continuously adapts schedules in response to unpredictable human behavior, maintaining both robustness and efficiency. We validate HERON on long-horizon collaboration tasks consisting of multiple interdependent sub-tasks that cannot be solved purely in parallel due to strong temporal dependencies. Our evaluation assesses both task completion accuracy and overall execution time, demonstrating the framework's ability to deliver resilient and efficient HRC under stochastic human uncertainties such as performance variability, interruptions, and goal changes.

The main contributions of this paper are:

- **Human Uncertainty-Aware Scheduling Framework**: We introduce a scheduling framework that accounts for human variability, unexpected interruptions, and dynamic goal changes, and supports real-time adjustments to preserve robustness and resilience in long-horizon human–robot collaboration.

- **Physics-Guided LLM**: We incorporate robot kinematic and actuation constraints into the LLM-based estimation of execution times, enabling task assignments that remain physically valid while improving overall efficiency.

- **Integration of LLMs with Optimization**: We present a unified method that merges LLM-driven task decomposition with a MILP-based optimizer, delivering an adaptive and generalizable solution for efficient long-horizon human–robot collaboration beyond the capabilities of purely LLM-driven planners.

## 2 RELATED WORK

**LMs for Long-Horizon Task Planning** Language models (LMs) have been increasingly explored for long-horizon task planning, where complex goals must be decomposed into interdependent sub-tasks and executed under temporal and physical constraints. Among the earliest attempts, SMART-LLM introduced a structured pipeline for multi-robot planning that used LLMs for task decomposition, coalition formation, and allocation, demonstrating the feasibility of programmatic prompting to generate executable plans in both simulation and real-world scenarios (Kannan et al., 2024). Building on this foundation, subsequent works have addressed key limitations by incorporating grounding, formalization, and coordination mechanisms. Vision-language approaches such as VLM-TAMP link high-level semantic reasoning with task and motion planning to ensure physical feasibility in manipulation tasks (Yang et al., 2025b). Pipelines like Cook2LTL formalize natural language into temporal logic formulae for robotic execution, improving reliability and reproducibility in long-horizon tasks (Mavrogiannis et al., 2024). For multi-agent settings, frameworks such as LaMMA-P integrate

LLM-driven PDDL generation with symbolic planners to achieve more generalizable team planning (Zhang et al., 2025), while LLaMAR introduces iterative plan–act–correct–verify loops to enhance adaptability in partially observable environments (Nayak et al., 2024). Cooperative approaches like CaPo optimize multi-agent efficiency through meta-plan refinement and dynamic coordination (Liu et al., 2024), and Agent-Oriented Planning emphasizes decomposition principles such as solvability and non-redundancy to improve collaboration structure (Li et al., 2024a). However, despite these advances, existing LLM-based approaches often produce verbose or redundant plans, lack mechanisms to optimize resource and time efficiency, and struggle to incorporate domain-specific constraints such as physical feasibility or timing.

**LMs for Human-Robot Collaborations** Bringing long-horizon planning into HRC introduces unique challenges, as humans exhibit variability in execution times, interruptions, and shifting goals. Approaches that rely solely on smaller LMs, do not employ LLMs, or only make limited use of them—such as multimodal and hierarchical frameworks that integrate vision and speech cues—improve robustness in long-term assembly tasks and enhance user satisfaction (Yu et al., 2024), but they rely heavily on engineered pipelines and remain difficult to generalize across diverse collaboration settings (Ahn et al., 2024; Skreta et al., 2024; Shi et al., 2024; Fan & Zheng, 2024). Some studies apply LLMs directly to adaptive collaboration: PlanCollabNL translates natural language goals into structured plans that account for human conditions and dynamically allocate tasks (Izquierdo-Badiola et al., 2024), adaptive plan generation frameworks refine subgoals online based on human behavior, and intention-tracking systems like LIT allow robots to proactively assist in cooking scenarios by predicting human goals (Huang et al., 2024). Recent works also explore natural language-based instruction for collaborative assembly (Gao et al., 2024; Asuzu et al., 2025), humanoid cognition for proactive HRC (Li et al., 2024b), and state space exploration for adaptive guidance in collaborative mechanical assembly (Bashir et al., 2024; Lim et al., 2024; Hua et al., 2025). While these works illustrate the potential of LLM-based reasoning for HRC, they often cast robots in narrowly defined supportive roles—reacting to human instructions or intentions rather than engaging in joint, adaptive long-horizon planning. As a result, existing approaches are limited either by poor generalization or by restricting robots to local, supportive functions, leaving open the need for frameworks that enable balanced, resilient, and generalizable collaboration.

## 3 Preliminaries

**Human Uncertainty** In our study, the human collaborator is assumed to be capable of executing sub-tasks and providing basic feedback on their progress (e.g., for performance checks), but not of performing effective scheduling, and is modeled as an inherently uncertain agent whose variability must be accommodated by the system. Handling such uncertainty is thus a central challenge in HRC. To evaluate the robustness and resilience of HRC scheduling systems, we adopt simulation-based modeling, which is more practical and scalable than costly and low-repeatability user studies (Glogowski et al., 2017; Wang et al., 2022). Rather than replicating full cognitive or psychological states, our modeling focuses on computationally representing the core sources of uncertainty that significantly affect scheduling. Based on prior work in HRC (Ajoudani et al., 2018; Alirezazadeh & Alexandre, 2022; Liu et al., 2018), we categorize human uncertainty into three primary types, denoted as $\xi_1, \xi_2, \xi_3$, and design simulation models accordingly:

- *Performance Variability* ($\xi_1$): Even for identical tasks, the actual execution time may vary depending on individual skill, fatigue, or momentary condition. We model the execution time of a task $t$ as $T(t) \sim \mathcal{N}(\mu(t), \sigma^2)$, where $\mu(t)$ is the Estimated Execution Time (EET) predicted by the physics-guided LLM and $\sigma$ represents performance fluctuations.

- *Unexpected Behavior (Interruptions)* ($\xi_2$): Humans may suspend or resume tasks unpredictably due to distraction, external events, or personal reasons. We model this with an interruption indicator $I \sim$ Bernoulli($p$), and if an interruption occurs, its duration is modeled as $D \sim$ Uniform($d_{\min}, d_{\max}$), where $p$ is the interruption probability and $[d_{\min}, d_{\max}]$ specifies the range of suspension durations.

- *Dynamic Goal Changes* ($\xi_3$): Human preferences or intentions may change during execution, leading to modification of the overall objective or the required sub-tasks. We model such goal-change events as $G \sim$ Bernoulli($q$), where $q$ denotes the probability that a goal-change event occurs (e.g., preference shift or responding to an unexpected emergency).

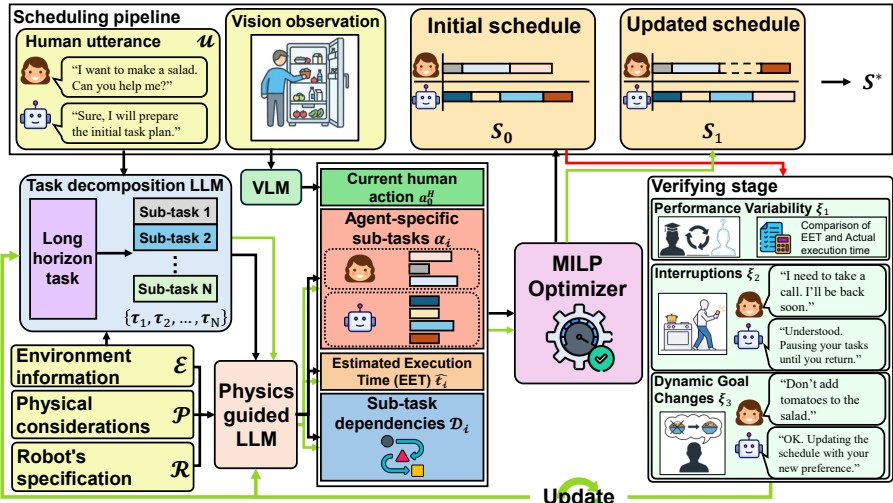

Figure 1: An overview of HERON's framework. Human utterances serve as inputs to the Task decomposition LLM, which identifies the long-horizon task and generates sub-tasks using environment information. These sub-tasks, combined with robot specifications and physical considerations, are processed by the Physics-guided LLM to produce agent-specific assignments, execution time estimates, and sub-task dependencies. The MILP Optimizer then outputs an initial schedule, which is iteratively verified and updated during execution to adapt to human uncertainty.

**Problem Definition** We formalize the problem of uncertainty-aware long-horizon scheduling for HRC as follows. Given a natural language utterance $u$ and the current human action observed via a vision-language model (VLM) $a_0^H$, the goal is to generate and maintain a feasible and efficient schedule $\mathcal{S}$ that assigns temporally ordered sub-tasks to human and robot agents, while adapting to human uncertainties during execution. First, the task decomposition LLM maps the human utterance and environmental information $\mathcal{E}$ into a set of sub-tasks $\mathcal{T} = \{\tau_1, \tau_2, \ldots, \tau_N\} = f_{\text{TD-LLM}}(u, \mathcal{E})$, where each sub-task $\tau_i$ is associated with preconditions and dependencies $\mathcal{D}_i$. Each sub-task is then enriched with EET and agent assignments using a physics-guided LLM, such that $(\hat{t}_i, \alpha_i) = f_{\text{P-LLM}}(\tau_i, \mathcal{E}, \mathcal{R}, \mathcal{P})$, where $\hat{t}_i$ denotes the EET and $\alpha_i \in \{H, R\}$ specifies assignment to the human (H) or robot (R), given robot specifications $\mathcal{R}$, physical considerations $\mathcal{P}$, and $\mathcal{E}$. $f_{\text{MILP}}$ then solves for an initial allocation and ordering of tasks as $\mathcal{S}_0 = \arg\min_{\mathcal{S}} C(\mathcal{S})$ s.t. $\text{Deps}(\mathcal{S}) \wedge \text{Feas}(\mathcal{S})$, where $C(\mathcal{S})$ is a cost function balancing efficiency and utilization, $\text{Deps}(\mathcal{S})$ encodes dependency constraints, and $\text{Feas}(\mathcal{S})$ enforces feasibility with respect to agent assignments and physical constraints. During execution, human uncertainties may arise, categorized as performance variability, unexpected behavior, or dynamic goal changes. Formally, let $\xi \in \{\xi_1, \xi_2, \xi_3\}$ represent uncertainty events; upon detecting $\xi$, the system re-enters the planning loop as $\mathcal{S}_{k+1} = f_{\text{MILP}}(f_{\text{P-LLM}}(f_{\text{TD-LLM}}(u', \mathcal{E}), \mathcal{R}))$, where $u'$ is either the original utterance or a new/modified instruction from the human, yielding an updated schedule $\mathcal{S}_{k+1}$. This iterative process converges to a stable schedule $\mathcal{S}^*$ after a finite number of updates, as the set of sub-tasks and uncertainty events is finite.

## 4 METHODOLOGY

Figure 1 illustrates the overall architecture of HERON, which integrates task decomposition, physics-guided reasoning, and optimization into a unified loop for uncertainty-aware HRC. The system takes two human-centered inputs: a natural language utterance $u$ describing the intended high-level task, and the current human action $a_0^H$ observed through a VLM. Along with environmental information $\mathcal{E}$, robot specifications $\mathcal{R}$, and physical considerations $\mathcal{P}$, these inputs are progressively processed by three core modules, resulting in an initial schedule $\mathcal{S}_0$ that assigns temporally ordered sub-tasks to human and robot agents. During execution, HERON continuously monitors for uncertainty events $\xi$, corresponding to performance variability, unexpected behavior, and dynamic goal

changes. These events are detected either by comparing observed human performance against EETs or by processing additional human utterances. When an event is triggered, the framework re-enters the planning loop, producing an updated schedule $\mathcal{S}_{k+1}$. This iterative verification and updating process ensures that HERON maintains efficiency, feasibility, and resilience in long-horizon HRC.

**Task decomposition LLM**  The Task decomposition LLM provides the entry point of the HERON pipeline by mapping a natural language utterance $u$ and environmental information $\mathcal{E}$ into a structured representation of sub-tasks. Formally, the module outputs a task graph $\mathcal{G} = (\mathcal{T}, \mathcal{D})$, where $\mathcal{T} = \{\tau_1, \tau_2, \ldots, \tau_N\} = f_{\text{TD-LLM}}(u, \mathcal{E})$. Each node $\tau_i$ corresponds to a sub-task and each directed edge $(\tau_i, \tau_j) \in \mathcal{D}$ denotes a dependency such that $\tau_i$ must be completed before $\tau_j$. This dependency structure can be represented as an adjacency matrix $A \in \{0, 1\}^{N \times N}$ with entries $A_{ij} = 1$ if $(\tau_i, \tau_j) \in \mathcal{D}$, and $A_{ij} = 0$ otherwise. In addition to symbolic task decomposition, the module grounds each sub-task to the environment by generating location mappings. Given a scene description $\mathcal{E}$, the function $\phi : \mathcal{T} \to \mathcal{L}$ assigns each $\tau_i$ to a start and end location, enabling subsequent modules to account for navigation costs. For instance, a task $\tau = \text{put\_Tomato\_in\_Bowl}$ is grounded as $\phi(\tau) = (l_{\text{Tomato}}, l_{\text{Bowl}})$. Thus, the output of this stage is a task graph $\mathcal{G}$ enriched with dependency relations $\mathcal{D}$ and environment-grounded location mappings $\phi(\tau)$, which is then passed to the Physics-guided LLM. In generating the task graph, the LLM occasionally introduces slightly conservative dependency relations, sometimes adding more ordering constraints than strictly required. This behavior is common in LLM-based parsing and tends to arise in long-horizon tasks that combine parallel and dependent subtasks, such as the cooking scenarios used in our evaluation. These conservative edges do not affect the correctness of the workflow, and the downstream MILP scheduler is able to recover feasible parallelism when appropriate. Also, the module operates under the assumption that the scene description $\mathcal{E}$ provides structured symbolic information, which supports consistent task grounding within our controlled simulation environment, though obtaining equivalent information from raw visual or multimodal inputs remains a plausible direction. Examples of the instruction prompts provided to the LLM and the corresponding decomposition outputs are shown in Appendix D.

**Physics-guided LLM**  The Physics-guided LLM augments the symbolic task graph $\mathcal{G} = (\mathcal{T}, \mathcal{D})$ with quantitative estimates and agent assignments. For each sub-task $\tau_i \in \mathcal{T}$, the module outputs both an EET $\hat{t}_i$ and an assignment label $\alpha_i \in \{H, R\}$, where $H$ denotes the human and $R$ denotes the robot, such that $(\hat{t}_i, \alpha_i) = f_{\text{P-LLM}}(\tau_i, \mathcal{E}, \mathcal{R}, \mathcal{P})$. First, feasibility is checked through a capability mapping $\kappa : \mathcal{T} \times \{H, R\} \to \{0, 1\}$, which prunes assignments that are physically impossible (e.g., $\kappa(\text{slice\_Tomato}, H) = 1, \kappa(\text{slice\_Tomato}, R) = 0$). For feasible cases, EET is estimated as $\hat{t}_i = t_i^{\text{mani}} + t_i^{\text{nav}}$, where $t_i^{\text{mani}}$ depends on kinematic and actuation constraints and $t_i^{\text{nav}}$ is derived from the environment-grounded mapping $\phi(\tau_i)$; for humans, $\hat{t}_i$ follows the stochastic distribution defined in Section 3. The final assignment is then formalized as $\alpha_i = \arg\min_{a \in \{H, R\}} \left[ \hat{t}_i(a) + \lambda \cdot C_{\text{switch}}(a) \right]$ subject to $\kappa(\tau_i, a) = 1$, where $C_{\text{switch}}$ penalizes frequent agent switching and $\lambda$ is a trade-off parameter. Importantly, this expression is not solved by a numeric optimizer but rather reflects how the Physics-guided LLM reasons over feasibility, execution efficiency, and coordination cost when deciding whether a sub-task is assigned to the human or the robot. The output of this stage is the enriched representation $\{(\tau_i, \hat{t}_i, \alpha_i)\}_{i=1}^{N}$ together with dependency structure $\mathcal{D}$, which forms the structured input to the MILP optimizer. Examples of the prompts given to the Physics-guided LLM and the resulting outputs are presented in Appendix E.

**MILP Optimizer**  The MILP optimizer takes as input the enriched representation from the Physics-guided LLM, including the task set $\mathcal{T}$, dependency relations $\mathcal{D}$, agent-specific execution times $\hat{t}_i(a)$, and feasibility constraints $\kappa(\tau_i, a)$. The decision variables consist of binary indicators $x_{i,a} \in \{0, 1\}$ for whether sub-task $\tau_i$ is assigned to agent $a \in \{H, R\}$, and continuous variables $s_i, c_i \in \mathbb{R}^+$ for task start and completion times. The optimization minimizes a cost function $C(\mathcal{S}) = \max_i c_i$. Constraints enforce that each task is assigned to exactly one feasible agent ($\sum_a x_{i,a} = 1$, $x_{i,a} \le \kappa(\tau_i, a)$), that completion times are consistent ($c_i = s_i + \sum_a x_{i,a} \hat{t}_i(a)$), that dependency relations are respected ($c_i \le s_j \ \forall (\tau_i, \tau_j) \in \mathcal{D}$), and that no agent executes overlapping tasks (disjunctive non-overlap constraints when $\alpha_i = \alpha_j$). Additional specifications, such as agent unavailability intervals or forced assignments, can be readily incorporated. Solving this MILP yields an initial feasible schedule $\mathcal{S}_0 = \{(\tau_i, \alpha_i, s_i, c_i)\}_{i=1}^{N}$, which specifies the assigned agent and

temporal allocation for every sub-task. More detailed formulations and pseudocode are provided in Appendix F.

**Verifying Stage**    After producing the initial schedule $\mathcal{S}_0$, HERON monitors execution and detects human uncertainties $\xi \in \{\xi_1, \xi_2, \xi_3\}$. For $\xi_1$ (performance variability), when the robot completes its first task the system queries the human via utterance to obtain feedback on the actual completion time $t_i^{\text{act}}$ of the parallel human sub-task. The system then compares this value with the predicted EET $\hat{t}_i$ and computes $\delta_i = |t_i^{\text{act}} - \hat{t}_i|$; if $\delta_i > \epsilon$ for a tolerance $\epsilon$, the remaining schedule is updated to reflect the new human execution rate. For $\xi_2$ (unexpected behavior) and $\xi_3$ (dynamic goal change), detection relies purely on verbal input from the human: interruptions or temporary unavailability trigger $\xi_2$, while new instructions that add, remove, or modify sub-tasks trigger $\xi_3$. In our implementation, this reliance on explicit verbal feedback is a deliberate design choice that keeps the monitoring process simple and reliable. Although more autonomous forms of monitoring utterance could be achieved through multimodal sensing such as foundation models, integrating such components lies beyond the scope of the current prototype and represents a natural direction for future extensions. When any event $\xi$ occurs, the framework re-enters the planning loop and updates the schedule as $\mathcal{S}_{k+1} = f_{\text{MILP}}(f_{\text{P-LLM}}(f_{\text{TD-LLM}}(u', \mathcal{E}), \mathcal{R}))$, where $u'$ is the most recent human utterance, and continues until no further events are raised, yielding a stable schedule $\mathcal{S}^*$. The process by which each type of uncertainty ($\xi_1, \xi_2, \xi_3$) is verified through experimental logs and leads to updated schedules can be examined task by task in Appendix G.

## 5 EXPERIMENTS

**Experimental Setup**    We conduct experiments in household kitchen scenarios, which are natural domains for evaluating HRC due to their complexity, temporal dependencies, and the need for mixed human and robot effort. The environment is implemented using the AI2-THOR simulator (Kolve et al., 2017), which provides a realistic, interactive kitchen setting. In practice, we extract the spatial coordinates of target objects in the simulator and use them to configure the environment as JSON files, which then serve as structured inputs to our code execution. Based on these environment settings, natural language instructions corresponding to the four task categories are provided to the LLMs, enabling them to reason over feasible task decompositions and agent assignments. We then compare HERON against alternative frameworks under these standardized environments. In addition, for a subset of representative cases, we directly deploy the generated schedules in the full 3D environment to validate that the robot can successfully execute the planned actions in practice. Inspired by household cooking scenarios, we evaluate four long-horizon collaborative tasks, each of which exhibits unique structural properties that together cover parallel, sequential, and hybrid dependencies, thereby allowing us to demonstrate HERON's ability to generalize scheduling to diverse HRC settings:

- *Task 1 - Salad preparation:* largely parallelizable subtasks such as cutting ingredients and placing them into a bowl; also used as a demonstration case.

- *Task 2 - Baking:* dominated by strong sequential dependencies arising from the repeated process of adding ingredients into a bowl and mixing batter.

- *Task 3 - Chicken with grilled vegetables:* a composite task combining parallel execution of two cooking processes, each of which internally contains sequential dependencies.

- *Task 4 - Pasta preparation:* requires both parallel tasks (e.g., boiling pasta while preparing sauce) and sequential steps (e.g., sautéing ingredients and combining them).

For each task, we run 20 independent trials. Each trial begins with an initial human utterance describing the task (selected from one of the four scenarios) and an observation of the human's current action, after which a stochastic human uncertainty module is activated to randomly introduce uncertainty events during execution. Given these inputs, HERON generates an initial schedule and dynamically updates it in response to observed uncertainties. The resulting schedule is executed within the AI2-THOR simulation to confirm feasibility, and in selected cases further validated in the full 3D environment. Evaluation metrics are described in the following paragraph.

**Metrics** Our evaluation metrics build upon four indicators inspired by LLaMAR (Nayak et al., 2024), with an additional metric introduced to capture the efficiency of schedule generation. We consider the following five metrics:

- *Success Rate (SR).* The fraction of trials in which the final task goal is achieved without violating dependency constraints, computed as $SR = \frac{\#\text{successful runs}}{\#\text{total runs}}$.

- *Transport Rate (TR).* The fraction of sub-tasks that are successfully executed within a trial, defined as $TR = \frac{\#\text{sub-tasks executed}}{\#\text{total sub-tasks}}$. This metric reflects how much of the long-horizon task is completed before failure or interruption, thereby measuring partial progress in addition to full success.

- *Balance (B).* The workload distribution between human and robot agents, measured as $Balance = \frac{\min(T_H, T_R)}{\max(T_H, T_R)}$, where $T_H$ and $T_R$ denote the cumulative durations assigned to the human and robot, respectively; values closer to 1 indicate more balanced collaboration.

- *Total Steps (TS).* The number of primitive actions executed by both agents, defined as $Steps = \sum_{i=1}^{N} \text{len}(\pi_i)$, where $\pi_i$ is the primitive action sequence for sub-task $\tau_i$; this metric serves as a proxy for plan compactness.

- *Timespan (TI).* The total execution time is computed using reference durations obtained from mobile manipulation datasets for all primitive sub-tasks. These dataset-based values are applied uniformly across HERON, baselines, and ablations, ensuring a fair comparison of scheduling efficiency. HERON still uses its EET predictions internally for scheduling, but TI is evaluated using the reference durations summarized in Appendix H.

**Baselines and Ablation Studies** For fair comparison, we benchmark HERON against recent LLM-based planning frameworks. Since HERON operates under the assumption that complete environment information (e.g., noiseless object positions and attributes) is available at execution time, we also provide the same perfect environment inputs to all baselines, even in cases where the original implementations assumed partial observability or direct simulator access. This ensures that all methods are evaluated under identical conditions.

- *SMART-LLM* (Kannan et al., 2024): A planning framework that augments LLM reasoning with structured environment information. SMART-LLM performs task decomposition based on language prompts and executes scheduling without explicit optimization.

- *Lip-LLM* (Obata et al., 2024): A framework that integrates LLM-based reasoning with external constraints to guide task allocation. Lip-LLM supports optimization through heuristic constraints.

- *LLaMAR* (Nayak et al., 2024): A recent framework designed for long-horizon multi-agent planning that leverages modular prompting. LLaMAR performs decomposition and reasoning based on local observations. However, for fair comparison, we modify the code to directly provide the same environment information used by HERON.

To analyze the contribution of each component of HERON, we conduct ablations by removing key modules:

- *w/o Physics-guided LLM:* In this setting, the framework does not employ the physics-guided module. Instead, the LLM is directly prompted to generate agent assignments using only common-sense reasoning, without structured physical constraints. This setting demonstrates the importance of physics-aware reasoning for producing feasible assignments and maintaining robustness.

- *w/o MILP Optimizer:* In this variant, the MILP optimizer is removed and the LLM alone produces task-to-agent assignments and scheduling decisions. This setting demonstrates the importance of explicit optimization for generating efficient and feasible schedules.

## 6 RESULTS AND DISCUSSION

**Quantitative Analysis** Table 1 summarizes results on four representative long-horizon HRC tasks. The collective metrics, including SR, TR, B, TS, and TI, show how each task's internal structure shapes overall collaboration efficiency. Tasks with a high degree of parallelism, such as Task 1, achieve both short TI and high TR while keeping B relatively even between agents. As sequential

Table 1: Comparison of HERON with baseline frameworks and ablations on Tasks 1–4.

| Algorithm | LLM | Task 1 (Salad preparation) | | | | | Task 2 (Baking) | | | | |
| | | SR | TR | B | TS | TI | SR | TR | B | TS | TI |
|---|---|---|---|---|---|---|---|---|---|---|---|
| SMART-LLM | GPT-5 | 0.25 | 0.53 | 0.70 | 7.75 | 99.31 | 0.15 | 0.43 | 0.80 | 6.50 | 164.19 |
| LiP-LLM | GPT-5 | 0.25 | 0.45 | 0.62 | 6.75 | 98.66 | 0.35 | 0.52 | 0.70 | 10.50 | 161.61 |
| LLaMAR | GPT-5 | 0.90 | 0.93 | 0.75 | 8.25 | 95.28 | 0.50 | 0.68 | 0.80 | 13.50 | 162.94 |
| **HERON** | **GPT-5** | **0.95** | **0.98** | **0.66** | **8.60** | **84.84** | **0.60** | **0.67** | **0.28** | **14.67** | **137.95** |
| HERON | Gemini-2.5-Pro | 0.80 | 0.80 | 0.53 | 9.50 | 97.33 | 0.60 | 0.60 | 0.28 | 17.67 | 193.82 |
| HERON w/o-pLLM | GPT-5 | 0.50 | 0.74 | 0.70 | 6.50 | 92.07 | 0.00 | 0.70 | 0.32 | 12.50 | - |
| HERON w/o-MILP | GPT-5 | 0.95 | 0.95 | 0.70 | 8.50 | 93.17 | 0.50 | 0.72 | 0.83 | 15.50 | 170.84 |
| Algorithm | LLM | Task 3 (Chicken with grilled vegetables) | | | | | Task 4 (Pasta preparation) | | | | |
| | | SR | TR | B | TS | TI | SR | TR | B | TS | TI |
| SMART-LLM | GPT-5 | 0.00 | 0.22 | 0.74 | 3.50 | - | 0.00 | 0.30 | 0.85 | 4.50 | - |
| LiP-LLM | GPT-5 | 0.00 | 0.32 | 0.65 | 6.50 | - | 0.00 | 0.50 | 0.52 | 7.25 | - |
| LLaMAR | GPT-5 | 0.80 | 0.82 | 0.80 | 17.25 | 1602.74 | 0.80 | 0.87 | 0.75 | 12.20 | 761.59 |
| **HERON** | **GPT-5** | **0.80** | **0.86** | **0.56** | **16.50** | **1511.47** | **0.80** | **0.98** | **0.63** | **13.00** | **700.88** |
| HERON | Gemini-2.5-Pro | 0.60 | 0.67 | 0.37 | 15.00 | 1446.37 | 0.60 | 0.68 | 0.40 | 17.00 | 1371.32 |
| HERON w/o-pLLM | GPT-5 | 0.50 | 0.73 | 0.29 | 18.50 | 1540.54 | 0.00 | 0.77 | 0.62 | 18.50 | - |
| HERON w/o-MILP | GPT-5 | 0.70 | 0.81 | 0.84 | 17.50 | 1615.10 | 0.80 | 0.98 | 0.89 | 13.00 | 751.96 |

Table 2: MILP re-planning average time (seconds) for each task and update count.

| Tasks | Number of Re-planning ($k$) | | |
| | 0 | 1 | 2 |
|---|---|---|---|
| Task 1 (Salad preparation) | 4.33 | 3.71 | 3.56 |
| Task 2 (Baking) | 5.59 | 5.29 | 5.26 |
| Task 3 (Chicken with grilled vegetables) | 5.11 | 4.75 | 4.40 |
| Task 4 (Pasta preparation) | 4.67 | 4.55 | 4.05 |

dependencies increase from Task 2 through Task 4, SR for robust planners remains competitive, yet TR steadily decreases and B declines, indicating that one agent is frequently forced to wait while the other completes prerequisite actions. Notably, Task 2 records lower SR and TR than the longer composite Tasks 3 and 4. This task forms an almost purely sequential pipeline, so even minor human interruptions propagate through the entire schedule and reduce both SR and TR, whereas the composite tasks contain partially independent sub-sequences that allow HERON's dynamic rescheduling to recover from local delays. Although TS varies only slightly across tasks, the ordering and coordination requirements rather than the sheer number of sub-tasks are the primary drivers of execution time. These observations confirm that HERON's scheduling approach effectively maintains efficiency and resilience despite the growing structural complexity and human uncertainty present in the more challenging tasks.

We evaluated HERON with two LLMs, GPT-5(OpenAI, 2025) and Gemini-2.5-Pro(Comanici et al., 2025), under identical prompts and experimental settings. All other components, including the physics-guided LLM and MILP optimizer, were unchanged so that differences reflect only the underlying LM. Across the four tasks, GPT-5 consistently achieved higher SR and shorter TI. Gemini-2.5-Pro often produced overly constrained dependency structures that prevented feasible schedules, and during physics-guided reasoning it sometimes failed to recognize actions requiring uniquely human skills (e.g., using fire or fine-grained manipulation), leading to early termination or missed sub-tasks. These results highlight that planning performance varies with the LM. To ensure fairness, all subsequent experiments used GPT-5 as the default.

Notably, all four tasks in our evaluation were executed under simulated human uncertainty, and the consistently high SR and TR achieved by HERON therefore reflect its resilience to stochastic performance fluctuations, interruptions, and goal changes. Moreover, TI is computed using dataset-based execution durations derived from multiple mobile-manipulation datasets, and HERON's consistently

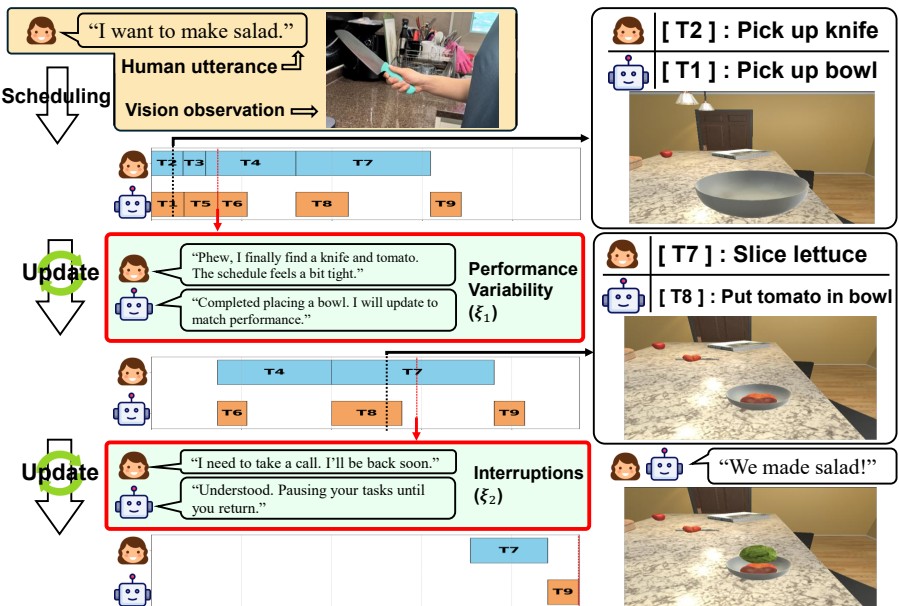

Figure 2: Dynamic scheduling timeline and recorded task trace for the Task 1 showing two re-planning events caused by $\xi_1$ and $\xi_2$. Sub-tasks are labeled **T1**–**T9** as follows: **T1** – pick up bowl, **T2** – pick up knife, **T3** – pick up tomato, **T4** – slice tomato, **T5** – put bowl on CounterTop, **T6** – pick up lettuce, **T7** – slice lettuce, **T8** – put tomato in bowl, **T9** – place lettuce in bowl.

shorter TI demonstrates its efficiency under a realistic timing model. This improvement further indicates that the physics-guided reasoning—by incorporating kinematic feasibility, actuation limits, and EET predictions—provides meaningful structural advantages that translate into more effective scheduling. Static planners such as SMART-LLM exhibit uniformly low SR and long TI because they cannot adapt to stochastic human interruptions and lack any optimizer to minimize idle time. LiP-LLM benefits from a simple linear-programming scheduler, which yields moderate improvements in TR and TI, but it still struggles when unexpected delays require dynamic reallocation. LLaMAR performs substantially better than these static methods, reflecting the advantage of dynamic planning, yet its absence of a formal optimizer such as MILP leads to longer TI and less balanced workloads compared with HERON. These trends confirm that dynamic re-scheduling alone is not sufficient: coupling a robust planner with an optimizer is critical for efficient long-horizon human–robot collaboration.

Ablation results demonstrate the importance of HERON's core components. Removing the physics-guided LLM (pLLM) causes a sharp decline in SR because the system can no longer recognize actions that require uniquely human skills such as precise manipulation or operations involving heat. Eliminating the MILP optimizer forces a simple first-in-first-out scheduling that slightly raises B but increases TI and lowers SR as human interruptions create idle periods. These findings confirm that both the physics-aware reasoning of the pLLM and the optimization of the MILP are essential to sustaining high success rates and efficient execution.

We also analyzed the computational overhead associated with dynamic re-planning. As summarized in Table 2, the MILP solving time ranges from approximately 3.56 to 5.59 seconds, and the detailed measurement procedure is provided in Appendix I. These runtimes are modest compared with the duration of individual sub-tasks in our long-horizon scenarios, which typically span tens of seconds. Accordingly, HERON can update schedules responsively without disrupting overall task progress, reinforcing its practical efficiency and resilience during extended human–robot collaboration.

**Qualitative Analysis** To illustrate HERON's dynamic collaboration capability, we examined a long-horizon task preparing a salad in a simulated kitchen environment. The initial optimal schedule assigned the robot to pick up a bowl, pick up lettuce, and eventually combine the ingredients while the human picked up a knife, sliced a tomato, and the lettuce later. At the beginning, the robot

collected the bowl and lettuce while the human retrieved the tomato and started slicing it. The human noted that locating the knife and tomato was taking longer than expected and remarked that the schedule felt tighter than planned. In response, the robot acknowledged the update and prepared to adjust the plan once it completed the current action. HERON performed a first dynamic re-planning, extending the remaining time for the slicing action and rearranging subsequent steps without interrupting the tasks already in progress. Work progressed according to this updated plan until a second uncertainty event when the human paused during the slicing the lettuce. HERON again triggered re-planning, calculating the remaining schedule so the robot could efficiently wait and then complete its final actions once slicing resumed. This qualitative trajectory shows that applying the schedule generated by HERON enables the human and robot to collaborate and successfully complete the long-horizon salad-making task even in the presence of human uncertainty. Figure 2 illustrates this process, presenting the two re-planning events and the revised allocation of sub-tasks.

## 7 CONCLUSION AND LIMITATIONS

We presented HERON, a framework for HRC that integrates task decomposition, physics-guided reasoning, and optimization-based scheduling to address long-horizon planning under human uncertainty. Across four household collaboration tasks with stochastic human behavior, HERON achieved substantially higher robustness and efficiency than existing LLM-based planning frameworks. In particular, averaged over all baselines, HERON improves success rate by 45% while reducing schedule timespan by 13%, demonstrating its ability to maintain both accuracy and temporal efficiency in uncertainty-aware long-horizon scheduling.

Despite these advantages, our framework has two key limitations. First, the human collaborator is modeled as an agent who can execute sub-tasks and provide basic feedback on personal progress but cannot perform effective scheduling. This assumption enables detection of performance variability ($\xi_1$). While VLMs could in principle automate this verification, exploring such VLM-based monitoring lies beyond the current scope and is a promising direction for future work. Second, our study focuses on one-to-one human–robot collaboration. Scaling HERON to multi-human and multi-robot scenarios would require additional mechanisms for coordination, resource contention, and communication, which we leave as future research. Third, our evaluation is conducted entirely in simulation. This controlled setting allows us to systematically analyze uncertainty and task structure without confounding perception noise or hardware constraints, but it also limits the extent to which HERON's resilience can be assessed in real-world human–robot collaboration. Integrating physical robotics platforms and human participants constitutes a natural next step toward validating the framework in practical settings. A detailed breakdown of validated versus unvalidated aspects of our contributions is provided in Appendix K.

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

# APPENDIX

## TABLE OF CONTENTS

## A  Terminology

We summarize key terminology and abbreviations specific to this paper for clarity.

**HERON** *Human-robot collaboration with Efficient and Resilient OptimizatioN for long-horizon planning*, the proposed framework integrating task decomposition, physics-guided reasoning, and MILP-based scheduling.

**Task Decomposition LLM** An LLM module that maps natural language utterances and environment information into structured sub-tasks with dependency relations.

**Physics-guided LLM** An LLM module that incorporates physical constraints (robot kinematics, actuation limits, environment geometry) to estimate *Estimated Execution Times* (EET) and generate human/robot task assignments.

**MILP Optimizer** The optimization module in HERON based on *Mixed-Integer Linear Programming (MILP)*. It computes efficient task schedules given enriched sub-task representations from the Physics-guided LLM. The optimizer ensures feasibility under dependency and resource constraints, minimizes makespan and idle time, and dynamically updates schedules in response to human uncertainty.

**Estimated Execution Time (EET)** The predicted time for completing a sub-task, generated by the physics-guided LLM for both human and robot agents.

**Schedule (S)** A temporally ordered assignment of sub-tasks to agents (human or robot), including start and completion times. Variants:

- $S_0$: Initial schedule output by the MILP optimizer.
- $S^*$: Stable schedule after iterative re-planning under uncertainty.

**Human Uncertainty ($\xi$)** Random factors influencing human execution. Categorized into three types:

- Performance Variability ($\xi_1$): Stochastic variation in task duration.
- Unexpected Behavior / Interruptions ($\xi_2$): Temporary halts in task execution.
- Dynamic Goal Changes ($\xi_3$): Shifts in human preference or task objectives.

## B  Environments

### B.1  Overview

To validate the schedules generated by HERON, we deploy them in the AI2-THOR interactive 3D simulation platform, which provides photo-realistic scenes and a rich set of manipulable objects for embodied AI research. Our tasks focus exclusively on kitchen environments. Among the official AI2-THOR floorplans, we select only the kitchen layouts and extract their pickable objects, receptacles, and all navigation targets with associated coordinates for use in our experiments. In particular, for *Chicken with grilled vegetables* (Task 3) and *Pasta preparation* (Task 4) scenarios, some required ingredients are not natively available in the simulator. To represent these tasks, we modify the scene metadata so that visually and physically similar items can serve as proxies. Specifically, objects with comparable geometry and affordances are relabeled in the scene information to match the required object names and types, allowing the agents to perceive and manipulate them as chicken or pasta components. Figure 3 shows a representative view of the photorealistic AI2-THOR kitchen environment employed for simulation and validation.

### B.2  Action Space

For validation of HERON's generated schedules, we used the standard AI2-THOR action space, which provides low-level controls for navigation and object manipulation in the simulated kitchen environments. The following actions were available to the agents during all deployments.

**Navigation Action**

- MoveAhead: Moves the agent one fixed step forward in its current facing direction. (0.25m)
- RotateLeft: Rotates the agent's view to the left by a preset angle. (90 degrees)

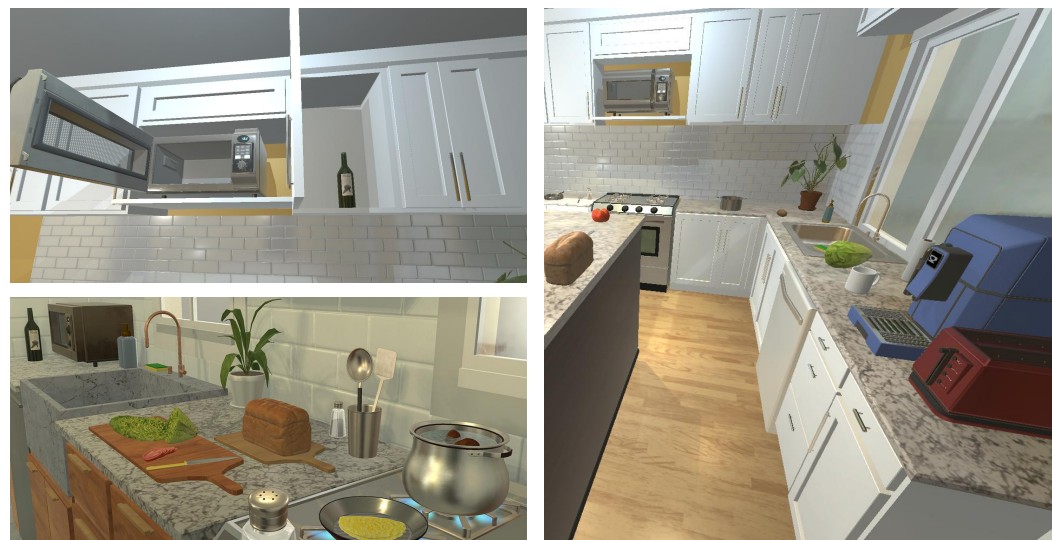

Figure 3: Photorealistic kitchen scene from the AI2-THOR simulator showing the environment used to validate HERON's generated schedules.

- `RotateRight`: Rotates the agent's view to the right by a preset angle. (90 degrees)
- `LookUp`: Tilts the camera upward to observe objects located higher in the scene. (15 degrees)
- `LookDown`: Tilts the camera downward to inspect objects or surfaces closer to the floor. (15 degrees)
- `GotoLocation(<location>)`: Navigates directly to a specified coordinate or predefined waypoint within the environment.

**Interaction Action**

- `PickupObject(<objectID>)`: Picks up the specified object and holds it in the agent's hand.
- `PutObject(<objectID>,<receptacleObjectID>)`: Places the held object into or onto the designated receptacle, such as a bowl, countertop, or cabinet.
- `OpenObject(<objectID>)`: Opens a container or door-like object, such as a cabinet, drawer, or refrigerator.
- `CloseObject(<objectID>)`: Closes an object that was previously opened.
- `SliceObject(<objectID>)`: Slices the specified object when an appropriate tool and supporting surface are available.
- `ToggleObjectOn(<objectID>)`: Turns on a powered object, such as a stove burner or microwave.
- `ToggleObjectOff(<objectID>)`: Turns off a powered object that is currently on.

Note that to prevent an excessive number of sub-tasks and redundant low-level actions in the schedule, we include AI2-THOR's navigation action, `GotoLocation` as a high-level action in HERON. For example, a single sub-task such as pickup apple in HERON is represented as `GotoLocation(Apple's location)` followed by `PickupObject(Apple)`. During data generation and optimization, the planner provides the estimated manipulation time and the navigation time from the start to the target position, allowing the scheduler to search over navigation–manipulation combinations and select the path with minimal total duration. Also, cooking-related sub-tasks such as Heat or Boil in HERON planner are expressed as combinations of the high-level `GotoLocation` and interaction actions in the AI2-THOR simulator. Rather than defining special commands, the planner composes standard primitives: the agent first navigates to the target location using actions like `GotoLocation`, then performs the required manipulations such as `PickupObject`, `PutObject`, or `ToggleObjectOn/Off` to operate appliances or place ingredients.

## C  SIMULATED HUMAN UNCERTAINTIES

### C.1  OVERVIEW

To realistically evaluate resilience in HRC, our experiments explicitly incorporated simulated human uncertainties. These uncertainties reflect conditions that frequently arise in real-world collaborative settings, and were designed to stress-test the ability of HERON to adapt schedules dynamically. We highlight three primary sources of uncertainty:

- **Performance Variability.** Even when a human repeats the same sub-task, the actual time taken can vary due to differences in skill, momentary attention, or physical condition such as fatigue. For example, slicing a vegetable may sometimes take noticeably longer if the human is distracted or handling the object more carefully. In our simulation, this was modeled as variation in the duration of each task, so that schedules could not rely on a fixed execution time. By introducing such variability, we tested whether the system could recognize deviations and update the schedule accordingly.

- **Interruptions.** Humans may occasionally stop working altogether, not because of the task itself but because of external or personal factors. Typical examples include pausing to answer a phone call, leaving the room briefly, or attending to another activity before resuming. In the simulation, interruptions were modeled as temporary suspensions of task execution with uncertain duration. This forces the robot to wait or adjust its own tasks, thereby testing whether the framework can maintain efficiency in the face of temporary but unpredictable gaps in human activity.

- **Goal Changes.** Humans can change their minds about the overall objective or the details of how a task should be completed. For instance, after requesting a salad, the human might decide to exclude a certain ingredient, or midway through baking might ask to prepare a different side dish. In our setup, such events appeared as updates to the high-level goal or modifications to sub-task requirements. This type of uncertainty is particularly challenging because it requires not just adjusting timing but rethinking the actual plan structure. Including goal changes in the evaluation allowed us to test the system's capacity for flexible and adaptive re-planning rather than rigid execution.

Together, these three categories of simulated uncertainty create a more realistic and dynamic testing environment, ensuring that HERON is evaluated not only on static efficiency but also on its robustness to the unpredictable nature of human collaborators. The pseudocode of our human uncertainty simulator can be found in Algorithm

## C.2 PSEUDOCODE

---

**Algorithm 1** HERON–Human Uncertainty Simulator($\mathcal{H}, \theta, s, c, t_0,$ `gen_async`)

---

1: **Inputs:**
2:     Human schedule $\mathcal{H} = \{(\tau, \hat{s}(\tau), \hat{e}(\tau))\}$
3:     Randomizer params $\theta = (\mu, \sigma, m_{\min}, m_{\max}, p_{\text{pause}}, d_{\min}, d_{\max}, q_{\text{goal}}, o_{\min}, o_{\max})$
4:     Scenario mode $s \in \{\text{pause}, \text{goal\_change}, \text{none}\}$
5:     Calibration flag $c \in \{\text{true}, \text{false}\}$
6:     Simulation start time $t_0$; async flag `gen_async`$\in \{\text{true}, \text{false}\}$
7: **Output:**    Event log $\mathcal{E}$ of *human_task_actual_start/end*, *human_interrupt_pause*, *human_interrupt_goal_change* (sorted)
8:
9: **Initialization**
10: Draw global multiplier $\kappa \sim \text{TruncNorm}(\mu, \sigma, [m_{\min}, m_{\max}])$ *// Parameters from code: $\mu = 1.0, \sigma = 0.1, m_{\min} = 0.7, m_{\max} = 1.3$*
11: Sort $\mathcal{H}$ by planned start to get $\{(\tau_i, \hat{s}_i, \hat{e}_i)\}_{i=1}^N$
12: $\mathcal{E} \leftarrow \emptyset$;    goal_triggered $\leftarrow$ false;    pause_triggered $\leftarrow$ false
13: **if** `gen_async` = true **then**
14:     Set $trigger\_idx = 0$ if $s = \text{goal\_change}$ and $N > 0$; otherwise set $trigger\_idx = -1$.
15:     Set $p_{\text{pause}}^{\text{task}} = p_{\text{pause}}$ if $s = \text{pause}$; otherwise 0.
16:     Set $q_{\text{goal}}^{\text{eff}} = q_{\text{goal}}$ if $s = \text{goal\_change}$; otherwise 0.
17: **else**
18:     $trigger\_idx \leftarrow -1$;    $p_{\text{pause}}^{\text{task}} \leftarrow 0$;    $q_{\text{goal}}^{\text{eff}} \leftarrow 0$
19: **end if**
20: *// Parameters from code: $p_{pause} = 0.5$ (pause mode), $q_{goal} = 1.0$ (goal_change mode)*
21:
22: **for** $i = 1$ **to** $N$ **do**
23:     $planned\_dur \leftarrow \hat{e}_i - \hat{s}_i$
24:     $actual\_start \leftarrow \hat{s}_i$
25:     Append $(\tau_i, \text{human\_task\_actual\_start}, actual\_start)$ to $\mathcal{E}$
26:     *// Performance variability (multiplicative, global $\kappa$)*
27:     **if** $c = \text{true}$ **then** $base\_dur \leftarrow planned\_dur$ **else** $base\_dur \leftarrow \kappa \cdot planned\_dur$
28:     $pause\_dur \leftarrow 0$
29:     **if** $s = \text{pause}$ and not $pause\_triggered$ and $U(0,1) < p_{\text{pause}}^{\text{task}}$ **then**
30:         $pause\_triggered \leftarrow$ true
31:         *// Interruption: duration and timing sampled from variables*
32:         $pause\_dur \leftarrow U(d_{\min}, d_{\max})$ *// Parameters from code: $d_{\min} = 5$, $d_{\max} = 15$*
33:         $offset \leftarrow U(o_{\min}, o_{\max}) \cdot base\_dur$ *// Parameters from code: $o_{\min} = 0.3$, $o_{\max} = 0.7$*
34:         $pause\_time \leftarrow actual\_start + offset$
35:         Append $(\text{human\_interrupt\_pause}, pause\_time, \tau_i, \text{duration} = pause\_dur)$ to $\mathcal{E}$
36:     **end if**
37:     $actual\_end \leftarrow actual\_start + base\_dur + pause\_dur$
38:     Append $(\tau_i, \text{human\_task\_actual\_end}, actual\_end, \hat{s}_i, \hat{e}_i)$ to $\mathcal{E}$
39:     *// Goal change*
40:     **if** `gen_async`=true and $s = \text{goal\_change}$ and not $goal\_triggered$ and $(i - 1) = trigger\_idx$ **then**
41:         **if** $U(0,1) < q_{\text{goal}}^{\text{eff}}$ **then**
42:             $goal\_triggered \leftarrow$ true
43:             Append $(\text{human\_interrupt\_goal\_change}, actual\_end, \text{trigger\_task} = \tau_i)$ to $\mathcal{E}$
44:         **end if**
45:     **end if**
46: **end for**
47: **return** sort_by_time($\mathcal{E}$)

---

# D    TASK DECOMPOSITION LLM

## D.1    OVERVIEW

The **Task Decomposition LLM** is the entry module of HERON. Its role is to translate a natural language instruction from the human, together with basic environment information, into a set of well-structured sub-tasks. The output is not just a list of actions, but a task graph where sub-tasks are connected through their temporal and logical dependencies. In addition, each sub-task is grounded to specific objects and locations in the environment (e.g., "pick up the tomato from the counter" rather than just "pick up tomato"), so that downstream modules can reason about feasibility and navigation costs. This structured task representation forms the symbolic backbone of the pipeline and provides the foundation for physics-aware reasoning in the next stage.

## D.2    INSTRUCTION

---

### Prompt for Task Decomposition LLM

**# ROLE & GOAL:** You are an expert AI planner specializing in human-robot collaboration within a simulated kitchen environment. Your task is to decompose a high-level goal into a sequence of sub-tasks and define each dependency.

**# CONTEXT:** The JSON object defines the entire kitchen scene, including all available objects, their properties, and their initial locations. You must use objects and receptacles found in this JSON. You must perform the following actions:

[*"pickup_<object_type>"*, *"put_<object_type>_on_<receptacle_type>"*,
*"put_<object_type>_in_<receptacle_type>"*, *"open_<object_type>"*, *"close_<object_type>"*,
*"slice_<object_type>"*, *"Heat_<object_type>_by_<receptacle_type>"*,
*"Boil_<object_type>_by_<receptacle_type>"*]

Note that, the placeholders '*<object_type>*' and '*<receptacle_type>*' refer to a single, specific object or receptacle type. If a similar action is needed again (e.g., opening a microwave twice), append a number to distinguish it (e.g., '*open_Microwave_1*', '*open_Microwave_2*').

**# INSTRUCTION:**
The high-level goal is "to make a salad".
When you define the dependencies, follow the guidelines:

1. Identify Independent Branches: Analyze the goal to find sub-processes that can be completed independently. Structure these as separate dependency branches that can be executed simultaneously by different agents.
2. Minimize the Critical Path: Do NOT create a long, single chain of dependencies. It is important to make this sequence as short as possible by branching out all non-dependent tasks.
3. Delay Dependencies: Only create a dependency between two tasks if one is strictly required for the other to begin. Avoid creating dependencies between the parallel branches until the final combination step.

**# OUTPUT FORMAT:** The output must be a single, valid JSON object. Do not add any explanations or text outside of the JSON object. You MUST use the following keys and structure precisely:

---

```
(CONTINUED) Prompt for Task Decomposition LLM

{
    "tasks": [
        "sub_task_1",
        "sub_task_2",
        ...
    ],
    "dependencies": [
        ["sub_task_1", "sub_task_2"],
        ...
    ]
}

# EXAMPLE:
Goal: "Make a cup of coffee"
{
    "tasks": [
        "pickup_Mug",
        "put_Mug_in_CoffeeMachine",
        "toggle_on_CoffeeMachine",
        "toggle_off_CoffeeMachine"
    ],
    "dependencies": [
        ["pickup_Mug", "put_Mug_in_CoffeeMachine"],
        ["put_Mug_in_CoffeeMachine", "toggle_on_CoffeeMachine"],
        ["toggle_on_CoffeeMachine", "toggle_off_CoffeeMachine"]
    ]
}

* NOTE: DO NOT OUTPUT ANYTHING EXTRA OTHER THAN WHAT HAS BEEN
SPECIFIED
```

## D.3 PROMPT OUTPUT EXAMPLE

```
Output from Task Decomposition LLM

{
    "tasks": [
        "pickup_Knife",
        "pickup_Lettuce",
        "slice_Lettuce",
        "pickup_Tomato",
        "slice_Tomato",
        "pickup_Apple",
        "slice_Apple",
        "pickup_Bowl",
        "put_Lettuce_in_Bowl",
        "put_Tomato_in_Bowl",
        "put_Apple_in_Bowl"
    ],
```

---

**(CONTINUED) Output from Task Decomposition LLM**

```
    "dependencies": [
        ["pickup_Knife", "slice_Lettuce"],
        ["pickup_Lettuce", "slice_Lettuce"],
        ["pickup_Knife", "slice_Tomato"],
        ["pickup_Tomato", "slice_Tomato"],
        ["pickup_Knife", "slice_Apple"],
        ["pickup_Apple", "slice_Apple"],
        ["pickup_Bowl", "put_Lettuce_in_Bowl"],
        ["slice_Lettuce","put_Lettuce_in_Bowl"],
        ["pickup_Bowl", "put_Tomato_in_Bowl"],
        ["slice_Tomato", "put_Tomato_in_Bowl"],
        ["pickup_Bowl", "put_Apple_in_Bowl"],
        ["slice_Apple", "put_Apple_in_Bowl"]
    ]
}
```

---

# E   PHYSICS-GUIDED LLM

## E.1   OVERVIEW

The **Physics-guided LLM** enriches the symbolic task graph produced by the task decomposition LLM with two additional layers of information: estimated execution time (EET) and agent assignments. Unlike purely semantic planners, this module explicitly reasons about physical constraints, such as robot kinematics, manipulation limits, and distances between objects. For each sub-task, it estimates how long it would take for the human or the robot to perform it, and then decides which agent is more suitable. The assignments are made not only based on feasibility (e.g., only the human can use a knife, only the robot can reach certain shelves) but also on efficiency and coordination, penalizing excessive switching between agents. The result is a task graph that is both grounded in the environment and annotated with realistic execution times and agent roles, which can then be passed to the MILP Optimizer to generate a concrete schedule.

## E.2   INSTRUCTION

---

**Prompt for Physics-guided LLM**

**# ROLE & GOAL:**
You are a robotics assistant specializing in motion planning and task time estimation for a mobile manipulator. Your goal is to analyze a list of sub-tasks and a scene definition JSON to produce all realistically and conservatively estimated execution time and capability data for an optimizer.

**# CONTEXT:**
The JSON object defines the entire kitchen scene, including all available objects, their properties, and their initial locations. Given sub-tasks:

---

**(CONTINUED) Prompt for Physics-guided LLM**

```
[
    "pickup_Knife",
    "pickup_Lettuce",
    "slice_Lettuce",
    "pickup_Tomato",
    "slice_Tomato",
    "pickup_Apple",
    "slice_Apple",
    "pickup_Bowl",
    "put_Lettuce_in_Bowl",
    "put_Tomato_in_Bowl",
    "put_Apple_in_Bowl"
]
```

# INSTRUCTION: Follow these steps precisely to gather all the necessary data.

## 1. Identify All Necessary Locations
- For each sub-task, determine its primary location from the environmental information.
- Compile a list of all unique locations involved in the plan.

## 2. Estimate Time Components (Navigation and Manipulation)
- Based on the plan and locations, you will now estimate two distinct types of execution time:

### 2-A. Navigation Time:
- This is the time it takes for the robot's mobile base to travel between the locations
- Use the coordinates from "navigation_targets", the robot's specifications and physical considerations.
- This data will populate the 'nav_time' key in the JSON output.

### 2-B. Manipulation Time:
- This is the time it takes to perform the action at the location (e.g., arm movement, grasping, slicing, toggling).
- Estimate this for each sub-task in the plan using the robot's specifications and physical considerations.
- This data will populate the 'mani_time' key in the JSON output.

### 2-C. Cooking Time:
- For cooking-related actions like Heat or Boil, the time estimate is not only based on robot movement but also on the physical process of cooking.
- Use your commonsense knowledge to provide a realistic time estimate for the food to be cooked.

## 3: Determine Capability ('capability')
- For each sub-task, determine if the robot and human can perform it. This will populate the 'capability' key.
- You must realistically consider the robot's physical limitations such as human-level dexterity and complex judgment related to cooking.

**(CONTINUED) Prompt for Physics-guided LLM**

**# ROBOT SPECIFICATIONS**
- Mobile_Base (Clearpath Robotics's Husky):
    - max_linear_velocity: 1.0 m/s
    - max_angular_velocity: 120 deg/s
    - linear_acceleration: 0.5 m/s$^2$
- Manipulator_Arm (Universal Robots's UR3):
    - dof: 6-DOF
    - max_reach: 0.8 m
    - gripper_open_time: 0.7 s
    - gripper_close_time: 0.7 s
- Overheads:
    - navigation_overhead_time: 1.0 s (for perception and planning before moving)
    - manipulation_overhead_time: 2.0 s (for perception and fine-tuning before acting)

**# PHYSICAL CONSIDERATIONS**

## 1. Navigation
- Consider the phases of motion: acceleration, cruising, and deceleration. The robot may not reach its maximum speed if the travel distance is short.
- Account for both rotation and translation time, plus the specified navigation overhead.
- Rotation Time: $t$ = angle / angular_velocity (where angle is in degrees, and angular_velocity is in deg/s)
- Uniform Acceleration Motion: $v = v_0 + at$, $s = v_0t + 0.5at^2$ (where $s$ is distance, $v$ is final velocity, $v_0$ is initial velocity, $a$ is acceleration, and $t$ is time)
- Distance to Reach Max Velocity: To determine if the robot will have a cruising phase, you can calculate the minimum distance required to accelerate to max velocity ($v$) from rest using $s = v^2/(2a)$.

## 2. Manipulation
- Manipulation includes multiple stages: pre-shaping Gripper, arm movement to the target, fine-tuning the pose, gripper actuation (grasping or releasing), Secure Lift and retreating. Time estimate must account for all these phases.
- Include the specified manipulation_overhead_time for perception and planning.
- Arm Movement Time: Estimate the time for the arm to move by approximating the distance the end-effector travels and dividing by an average speed. $t = d/v_{avg}$ (where $d$ is the estimated travel distance of the end-effector. Since $v_{avg}$ is not specified, assume a reasonable, conservative value like 0.25 m/s.)
- Fixed Time Actions: Always add the specified time for gripper actuation (gripper_open_time or gripper_close_time).
- Fine-Tuning: Add a fixed time for the fine adjustment, such as 1.0 to 1.5 seconds.

**# OUTPUT FORMAT**
The output must be a single, valid JSON object. Do not add any explanations or text outside of the JSON object. You MUST use the following keys and structure precisely:

```
{
    "capability": {
        "task_name_1": { "robot": 1(Yes) or 0 (No), "human": 1(Yes) or 0 (No) },
        "task_name_2": { "robot": 1(Yes) or 0 (No), "human": 1(Yes) or 0 (No) },
        "...": "..."
    },
```

```
(CONTINUED) Prompt for Physics-guided LLM

    "mani_time": {
      "task_name_1": time(seconds),
      "task_name_2": time(seconds),
      "...": "..."
    },
    "nav_time": {
      "LocationA → LocationB": time(seconds),
      "LocationB → LocationC": time(seconds),
      "...": "..."
    }
}

# EXAMPLE
This is an example for a goal "Make a salad" with tasks ["Pickup_Lettuce", "Slice_Lettuce"]
and locations ["Fridge", "CounterTop"].

{
    "capability": {
      "Pickup_Lettuce": { "robot": 1, "human": 1 },
      "Slice_Lettuce": { "robot": 0, "human": 1 }
    },
    "mani_time": {
      "Pickup_Lettuce": 6.8,
      "Slice_Lettuce": 25.0
    },
    "nav_time": {
      "Fridge|-02.10|+00.00|+01.09 → CounterTop|-00.08|+01.15|00.00": 5.1,
      "CounterTop|-00.08|+01.15|00.00 → Fridge|-02.10|+00.00|+01.09": 5.1
    }
}

* NOTE: DO NOT OUTPUT ANYTHING EXTRA OTHER THAN WHAT HAS BEEN
SPECIFIED
```

### E.3 PROMPT OUTPUT EXAMPLE

```
Output from Physics-guided LLM

{
    "capability": {
      "pickup_Knife": { "robot": 1, "human": 1 },
      "pickup_Lettuce": { "robot": 1, "human": 1 },
      "slice_Lettuce": { "robot": 0, "human": 1 },
      "pickup_Tomato": { "robot": 1, "human": 1 },
      "slice_Tomato": { "robot": 0, "human": 1 },
      "pickup_Apple": { "robot": 1, "human": 1 },
      "slice_Apple": { "robot": 0, "human": 1 },
      "pickup_Bowl": { "robot": 1, "human": 1 },
      "put_Lettuce_in_Bowl": { "robot": 1, "human": 1 },
      "put_Tomato_in_Bowl": { "robot": 1, "human": 1 },
      "put_Apple_in_Bowl": { "robot": 1, "human": 1 }
    },
```

```
(CONTINUED) Output from Physics-guided LLM

    "mani_time": {
        "pickup_Knife": 6.4,
        "pickup_Lettuce": 6.6,
        "slice_Lettuce": 25.0,
        "pickup_Tomato": 6.0,
        "slice_Tomato": 25.0,
        "pickup_Apple": 6.0,
        "slice_Apple": 25.0,
        "pickup_Bowl": 6.0,
        "put_Lettuce_in_Bowl": 6.2,
        "put_Tomato_in_Bowl": 6.2,
        "put_Apple_in_Bowl": 6.2
    },

    "nav_time": {
        "Knife|-01.68|+00.79|-00.24 → Lettuce|-01.81|+00.98|-00.94": 2.5,
        "Lettuce|-01.81|+00.98|-00.94 → Tomato|-00.39|+01.14|-00.81": 3.5,
        "Tomato|-00.39|+01.14|-00.81 → Apple|-00.47|+01.15|+00.48": 4.0,
        "Apple|-00.47|+01.15|+00.48 → Bowl|+00.27|+01.10|-00.75": 4.5,
        "Bowl|+00.27|+01.10|-00.75 → CounterTop|-01.87|+00.95|-01.21": 4.5
    }
}
```

# F  MIXED-INTEGER LINEAR PROGRAMMING (MILP) OPTIMIZER

## F.1  OVERVIEW

The **MILP Optimizer** is the scheduling module in HERON that transforms the enriched task representation into an executable and time-efficient schedule. Given sub-tasks, precedence, agent-specific execution-time estimates, and feasibility constraints, it minimizes a weighted objective over makespan and workload while enforcing (i) single-agent assignment per sub-task, (ii) precedence (temporal) constraints, and (iii) per-agent non-overlap (resource) constraints. When a human-uncertainty event occurs, the optimizer is re-invoked with updated inputs (e.g., modified EET or dependencies, agent availability) and produces a revised schedule.

## F.2  INPUTS

- Tasks $\mathcal{T} = \{\tau_i\}_{i=1}^{N}$: set of sub-tasks.
- Agents $\mathcal{A} = \{H, R\}$: human and robot.
- Dependencies $\mathcal{D} \subseteq \mathcal{T} \times \mathcal{T}$: precedence relations.
- EET $\hat{t}_i(a) \in \mathbb{R}_{\geq 0}$: estimated execution time for task $\tau_i$ on agent $a$.
- Feasibility $\kappa(\tau_i, a) \in \{0, 1\}$: capability map for each $(\tau_i, a)$ pair.
- Big$-M$ $M \in \mathbb{R}_{\geq 0}$: large constant for disjunctive constraints.
- AgentConstraints (optional): per-agent fields, including
  - unavailable_until: start time offset if agent is initially unavailable.
  - (extendable: availability windows, switching penalties, etc.).
- WarmStart/Replan $S_k$ (optional): partial schedule from a previous run, with completed tasks fixed and updated inputs.

## F.3 PSEUDOCODE

---

**Algorithm 2** HERON–MILP-Optimizer($\mathcal{T}, \mathcal{D}, \hat{t}, \kappa, \texttt{AgentConstraints}, M, S_k$)

---

1: **Decision variables:**
2:    $x_{i,a} \in \{0,1\}$ // 1 if $\tau_i$ assigned to agent $a$
3:    $s_i, c_i \in \mathbb{R}_{\geq 0}$ // start and completion times of $\tau_i$
4:    $y_{i,j,a} \in \{0,1\}$ // ordering if $\tau_i$ and $\tau_j$ both on agent $a$
5:    $C_{\max} \in \mathbb{R}_{\geq 0}$ // makespan upper bound
6: **Objective:**
7:    Minimize $C_{\max}$
8: **Constraints:**
9:    *// Assignment and feasibility*
10: **for** $i = 1$ **to** $N$ **do**
11:    $\sum_{a \in \mathcal{A}} x_{i,a} = 1$
12:    $x_{i,a} \leq \kappa(\tau_i, a) \quad \forall a$
13: **end for**
14:    *// Timing and makespan*
15: **for** $i = 1$ **to** $N$ **do**
16:    $c_i = s_i + \sum_a x_{i,a} \hat{t}_i(a)$
17:    $C_{\max} \geq c_i$
18: **end for**
19:    *// Precedence (dependencies)*
20: **for all** $(\tau_i, \tau_j) \in \mathcal{D}$ **do**
21:    $c_i \leq s_j$
22: **end for**
23:    *// Agent non-overlap (disjunctive)*
24: **for all** $a \in \mathcal{A}$ **do**
25:    **for all** unordered pairs $\{i,j\}, i \neq j$ **do**
26:      $s_i \geq c_j - M(1 - y_{i,j,a})$
27:      $s_j \geq c_i - M y_{i,j,a}$
28:      $y_{i,j,a} \leq x_{i,a}, \quad y_{i,j,a} \leq x_{j,a}$
29:    **end for**
30: **end for**
31:    *// Agent availability (optional via AgentConstraints)*
32: **for all** $a \in \mathcal{A}$ with $\texttt{unavailable\_until} = u_a$ **do**
33:    **for all** $i = 1$ **to** $N$ **do**
34:      $s_i \geq u_a - M(1 - x_{i,a})$
35:    **end for**
36: **end for**
37:    *// Re-planning (optional)*
38: **if** $S_k$ provided **then**
39:    Fix completed tasks in $S_k$ ($x_{i,a}, s_i, c_i$); update $\hat{t}$ or $\mathcal{D}$ if changed; re-solve.
40: **end if**
41: **Solve** the MILP; **return** $S = \{(\tau_i, \alpha_i, s_i, c_i)\}$ with $\alpha_i = \arg\max_a x_{i,a}$.

---

# G   EXPERIMENTAL LOGS AND SCHEDULE VISUALIZATIONS

## G.1   LOGS AND VISUALIZATIONS FOR TASK 1

**Experimental Log for Task 1**

```
==================================================
STARTING NEW EXPERIMENT
Goal: 'to make a salad'
==================================================

— Initial Plan & Ground Truth —
— Optimal Schedule (from t=0.00s, Makespan: 87.80s) —
    - pickup_Knife | human | 0.00 → 6.40
    - pickup_Lettuce | robot | 0.00 → 6.60
    - slice_Lettuce | human | 6.60 → 31.60
    - pickup_Bowl | robot | 6.60 → 12.60
    - pickup_Tomato | robot | 19.60 → 25.60
    - pickup_Apple | robot | 25.60 → 31.60
    - slice_Apple | human | 31.60 → 56.60
    - put_Lettuce_in_Bowl | robot | 31.60 → 37.80
    - slice_Tomato | human | 56.60 → 81.60
    - put_Apple_in_Bowl | robot | 56.60 → 62.80
    - put_Tomato_in_Bowl | robot | 81.60 → 87.80

— Human's Ground Truth Behavior —
    - Task: pickup_Knife | Planned: 6.40s | Actual (GT): 7.08s | Actual End: 7.08s
    - Task: slice_Lettuce | Planned: 25.00s | Actual (GT): 27.65s | Actual End: 34.73s
    - Task: slice_Apple | Planned: 25.00s | Actual (GT): 27.65s | Actual End: 62.39s
    - Task: slice_Tomato | Planned: 25.00s | Actual (GT): 27.65s | Actual End: 90.04s
```

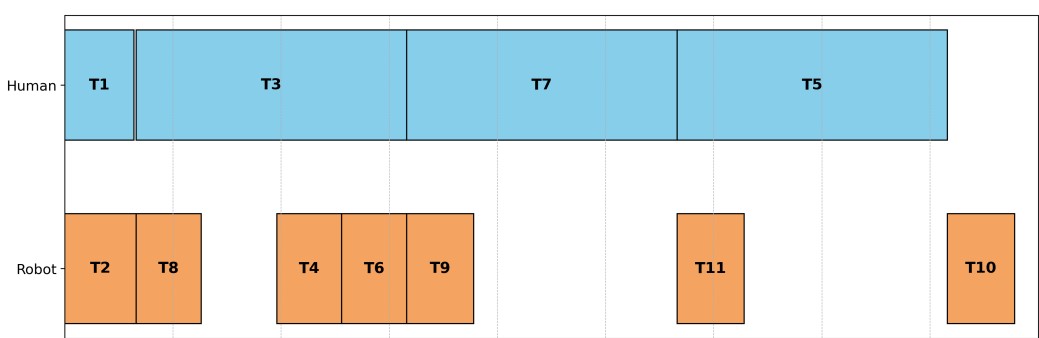

Figure 4: Initial schedule produced by HERON for making a salad (Task 1), showing parallel time-lines for the human (blue) and robot (orange) agents. Sub-tasks are labeled **T1**–**T11** as follows: **T1** – pick up knife, **T2** – pick up lettuce, **T3** – slice lettuce, **T4** – pick up tomato, **T5** – slice tomato, **T6** – pick up apple, **T7** – slice apple, **T8** – pickup bowl, **T9** – put lettuce in bowl, **T10** – put tomato in bowl, **T11** – put apple in bowl.

**Experimental Log for Task 1 (Goal Change)**

[SimManager @ t=7.08s] Event: 'human_interrupt_goal_change' on Task: 'pickup_Knife'
>>> INTERRUPT: Human requested a goal change: 'to make salad with tomato not apple'
— Triggering Dynamic Re-planning (Count: 1) —
    - Performing re-planning due to goal change.
    - New high-level goal: 'Make a salad with tomato not apple'

[Optimizer] Solution found (Status: OPTIMAL).
— Optimal Schedule (from t=7.08s, Makespan: 63.28s) —
    - slice_Lettuce | human | 7.08 → 32.08
    - pickup_Bowl | robot | 7.08 → 13.08
    - pickup_Tomato | robot | 13.08 → 19.08
    - slice_Tomato | human | 32.08 → 57.08
    - put_Lettuce_in_Bowl | robot | 57.08 → 63.28
    - put_Tomato_in_Bowl | human | 57.08 → 63.28

— Human's Ground Truth Behavior —
    - Task: slice_Lettuce | Planned: 25.00s | Actual (GT): 27.65s | Actual End: 34.73s
    - Task: slice_Tomato | Planned: 25.00s | Actual (GT): 27.65s | Actual End: 62.39s
    - Task: put_Tomato_in_Bowl | Planned: 6.20s | Actual (GT): 6.86s | Actual End: 69.25s

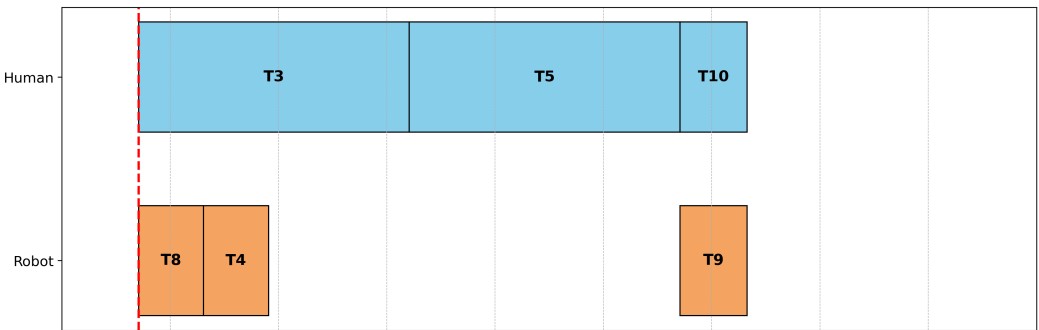

Figure 5: Updated schedule after the first re-planning caused by goal change. The human requested to make salad without apple and the schedule was recomputed to incorporate this new goal while preserving temporal consistency. Task labels follow the definitions given in Figure 4.

Experimental Log for Task 1 (Performance Variability)

[SimManager @ t=13.08s] Event: 'task_planned_end (robot)' on Task: 'pickup_Bowl'
>>> INTERRUPT: Human performance deviation detected.
— Triggering Dynamic Re-planning (Count: 2) —
    - Performing partial re-planning for remaining tasks.

[Optimizer] Solution found (Status: OPTIMAL).
— Optimal Schedule (from t=13.08s, Makespan: 68.59s) —
    - slice_Lettuce | human | 13.08 → 34.73
    - pickup_Tomato | robot | 28.73 → 34.73
    - slice_Tomato | human | 34.73 → 62.39
    - put_Lettuce_in_Bowl | robot | 34.73 → 40.93
    - put_Tomato_in_Bowl | robot | 62.39 → 68.59

— Human's Ground Truth Behavior —
    - Task: slice_Lettuce | Planned: 21.65s | Actual (GT): 21.65s | Actual End: 34.73s
    - Task: slice_Tomato | Planned: 27.65s | Actual (GT): 27.65s | Actual End: 62.39s

================================================
EXPERIMENT FINISHED
================================================

— FINAL RESULTS —

{
    "success": true,
    "final_makespan": 68.58872324461083,
    "replanning_count": 2,
    "transport_rate": 1.0,
    "balance": 0.6,
    "total_steps": 8
}

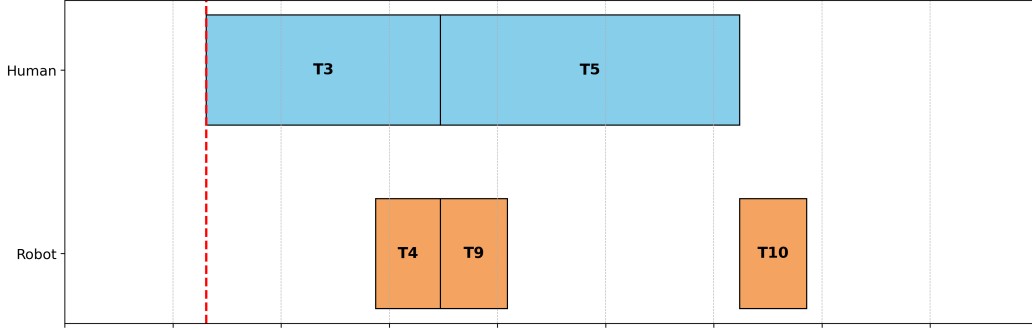

Figure 6: Updated schedule after the second re-planning triggered by performance deviation. Because the human's actual execution time exceeded the planned duration, the system classified the user as a novice and proportionally extended the remaining plan times to maintain overall feasibility. Task labels follow the definitions given in Figure 4.

## G.2 LOGS AND VISUALIZATIONS FOR TASK 2

**Experimental Log for Task 2**

```
==================================================
STARTING NEW EXPERIMENT
Goal: 'to bake a bread'
==================================================

— Initial Plan & Ground Truth —
— Optimal Schedule (from t=0.00s, Makespan: 140.00s) —
    - open_Fridge | robot | 0.00 → 4.00
    - pickup_Flour | robot | 4.00 → 11.00
    - pickup_Egg | human | 4.00 → 11.00
    - pickup_Mug_1 | robot | 11.00 → 18.00
    - close_Fridge | human | 14.00 → 18.00
    - pickup_Sugar | robot | 18.00 → 25.00
    - put_Flour_in_Mug | human | 18.00 → 26.00
    - pickup_SaltShaker | robot | 25.00 → 32.00
    - put_Sugar_in_Mug | human | 26.00 → 34.00
    - open_Microwave_1 | robot | 32.00 → 36.00
    - put_SaltShaker_in_Mug | human | 34.00 → 42.00
    - put_Egg_in_Mug | human | 42.00 → 50.00
    - put_Mug_in_Microwave | human | 50.00 → 58.00
    - close_Microwave | human | 58.00 → 62.00
    - Heat_Mug_by_Microwave | human | 62.00 → 122.00
    - open_Microwave_2 | human | 122.00 → 126.00
    - pickup_Mug_2 | human | 126.00 → 133.00
    - put_Mug_on_CounterTop | human | 133.00 → 140.00

— Human's Ground Truth Behavior —
    - Task: pickup_Egg | Planned: 7.00s | Actual (GT): 7.88s | Actual End: 11.88s
    - Task: close_Fridge | Planned: 4.00s | Actual (GT): 4.50s | Actual End: 18.50s
    - Task: put_Flour_in_Mug | Planned: 8.00s | Actual (GT): 9.00s | Actual End: 27.00s
    - Task: put_Sugar_in_Mug | Planned: 8.00s | Actual (GT): 9.00s | Actual End: 35.00s
    - Task: put_SaltShaker_in_Mug | Planned: 8.00s | Actual (GT): 9.00s | Actual End: 43.00s
    - Task: put_Egg_in_Mug | Planned: 8.00s | Actual (GT): 9.00s | Actual End: 51.00s
    - Task: put_Mug_in_Microwave | Planned: 8.00s | Actual (GT): 9.00s | Actual End: 59.00s
    - Task: close_Microwave | Planned: 4.00s | Actual (GT): 4.50s | Actual End: 62.50s
    - Task: open_Microwave_2 | Planned: 4.00s | Actual (GT): 4.50s | Actual End: 126.50s
    - Task: Heat_Mug_by_Microwave | Planned: 60.00s | Actual (GT): 67.50s | Actual End: 129.50s
    - Task: pickup_Mug_2 | Planned: 7.00s | Actual (GT): 7.88s | Actual End: 133.88s
    - Task: put_Mug_on_CounterTop | Planned: 7.00s | Actual (GT): 7.88s | Actual End: 140.88s
```

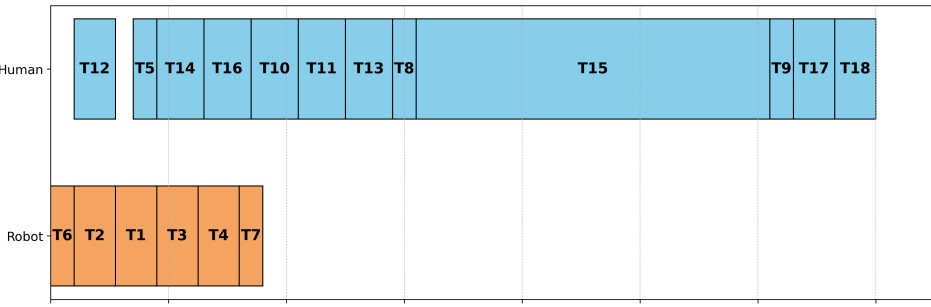

Figure 7: Initial schedule produced by HERON for baking a bread (Task 2), showing parallel timelines for the human (blue) and robot (orange) agents. Sub-tasks are labeled **T1**–**T18** as follows: **T1** – pick up mug 1, **T2** – pick up flour, **T3** – pick up sugar, **T4** – pick up saltshaker, **T5** – close fridge, **T6** – open fridge, **T7** – open microwave 1, **T8** – close microwave, **T9** – open microwave 2, **T10** – put saltshaker in mug, **T11** – put egg in mug, **T12** – pick up egg, **T13** – put mug in microwave, **T14** – put flour in mug, **T15** – heat mug by microwave, **T16** – put sugar in mug, **T17** – pick up mug 2, **T18** – put mug on countertop.

**Experimental Log for Task 2 (Performance Variability)**

[SimManager @ t=18.00s] Event: 'task_planned_end (robot)' on Task: 'pickup_Mug_1'
>>> INTERRUPT: Human performance deviation detected.
— Triggering Dynamic Re-planning (Count: 1) —
   - Performing partial re-planning for remaining tasks.

[Optimizer] Solution found (Status: FEASIBLE).
— Optimal Schedule (from t=18.00s, Makespan: 146.25s) —
   - pickup_Sugar | robot | 18.00 → 25.00
   - put_Flour_in_Mug | human | 18.00 → 27.00
   - close_Fridge | robot | 25.00 → 25.50
   - pickup_SaltShaker | robot | 25.50 → 40.00
   - put_Sugar_in_Mug | human | 27.00 → 36.00
   - put_Egg_in_Mug | human | 36.00 → 45.00
   - open_Microwave_1 | robot | 40.00 → 44.00
   - put_SaltShaker_in_Mug | robot | 44.00 → 52.00
   - put_Mug_in_Microwave | human | 45.00 → 54.00
   - close_Microwave | human | 54.00 → 58.50
   - Heat_Mug_by_Microwave | human | 58.50 → 126.00
   - open_Microwave_2 | human | 126.00 → 130.50
   - pickup_Mug_2 | human | 130.50 → 138.38
   - put_Mug_on_CounterTop | human | 138.38 → 146.25
— Human's Ground Truth Behavior —
   - Task: put_Flour_in_Mug | Planned: 9.00s | Actual (GT): 9.00s | Actual End: 27.00s
   - Task: put_Sugar_in_Mug | Planned: 9.00s | Actual (GT): 9.00s | Actual End: 36.00s
   - Task: put_Egg_in_Mug | Planned: 9.00s | Actual (GT): 9.00s | Actual End: 45.00s
   - Task: put_Mug_in_Microwave | Planned: 9.00s | Actual (GT): 9.00s | Actual End: 54.00s
   - Task: close_Microwave | Planned: 4.50s | Actual (GT): 4.50s | Actual End: 58.50s
   - Task: Heat_Mug_by_Microwave | Planned: 67.50s | Actual (GT): 67.50s | Actual End: 126.00s
   - Task: open_Microwave_2 | Planned: 4.50s | Actual (GT): 4.50s | Actual End: 130.50s
   - Task: pickup_Mug_2 | Planned: 7.88s | Actual (GT): 7.88s | Actual End: 138.38s
   - Task: put_Mug_on_CounterTop | Planned: 7.88s | Actual (GT): 7.88s | Actual End: 146.25s

====================================================
EXPERIMENT FINISHED
====================================================
— FINAL RESULTS —
{
    "success": true,
    "final_makespan": 146.25,
    "replanning_count": 1,
    "transport_rate": 1.0,
    "balance": 0.8,
    "total_steps": 18
}

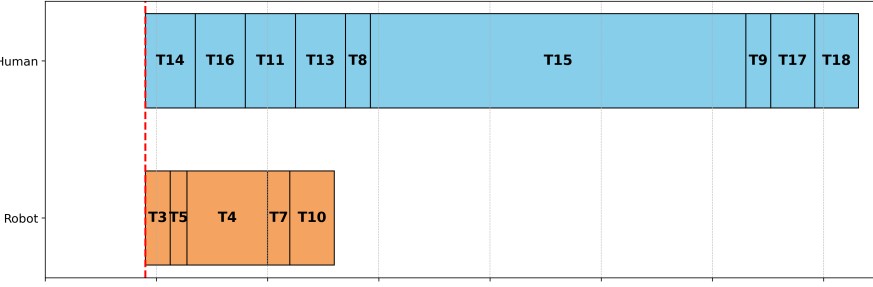

Figure 8: Updated schedule after the re-planning triggered by performance deviation. Because the human's actual execution time exceeded the planned duration, the system classified the user as a novice and proportionally extended the remaining plan times to maintain overall feasibility. Task labels follow the definitions given in Figure 7.

## G.3 LOGS AND VISUALIZATIONS FOR TASK 3

**Experimental Log for Task 3**

```
====================================================
STARTING NEW EXPERIMENT
Goal: 'to cook chicken with grilled vegetable'
====================================================

— Initial Plan & Ground Truth —
— Optimal Schedule (from t=0.00s, Makespan: 1613.20s) —
    - pickup_Pot | robot | 0.00 → 6.80
    - pickup_Chicken | robot | 6.80 → 13.60
    - put_Chicken_in_Pot | human | 13.60 → 20.40
    - open_Drawer | robot | 13.60 → 23.10
    - put_Pot_on_StoveBurner | human | 20.40 → 27.20
    - pickup_Potato | robot | 23.10 → 29.90
    - Heat_Pot_by_StoveBurner | human | 27.20 → 927.20
    - pickup_Tomato | robot | 29.90 → 42.10
    - pickup_Pan | robot | 42.10 → 48.90
    - pickup_Knife | human | 927.20 → 934.00
    - slice_Potato | human | 934.00 → 967.80
    - put_Potato_in_Pan | human | 967.80 → 974.60
    - slice_Tomato | human | 974.60 → 999.60
    - put_Tomato_in_Pan | human | 999.60 → 1006.40
    - put_Pan_on_StoveBurner | human | 1006.40 → 1013.20
    - Heat_Pan_by_StoveBurner | human | 1013.20 → 1613.20

— Human's Ground Truth Behavior —
    - Task: put_Chicken_in_Pot | Planned: 6.80s | Actual (GT): 5.89s | Actual End: 19.49s
    - Task: put_Pot_on_StoveBurner | Planned: 6.80s | Actual (GT): 5.89s | Actual End: 26.29s
    - Task: Heat_Pot_by_StoveBurner | Planned: 900.00s | Actual (GT): 779.22s | Actual End: 806.42s
    - Task: pickup_Knife | Planned: 6.80s | Actual (GT): 5.89s | Actual End: 933.09s
    - Task: slice_Potato | Planned: 33.80s | Actual (GT): 29.26s | Actual End: 963.26s
    - Task: put_Potato_in_Pan | Planned: 6.80s | Actual (GT): 5.89s | Actual End: 973.69s
    - Task: slice_Tomato | Planned: 25.00s | Actual (GT): 21.65s | Actual End: 996.25s
    - Task: put_Tomato_in_Pan | Planned: 6.80s | Actual (GT): 5.89s | Actual End: 1005.49s
    - Task: put_Pan_on_StoveBurner | Planned: 6.80s | Actual (GT): 5.89s | Actual End: 1012.29s
    - Task: Heat_Pan_by_StoveBurner | Planned: 600.00s | Actual (GT): 519.48s | Actual End: 1532.68s
```

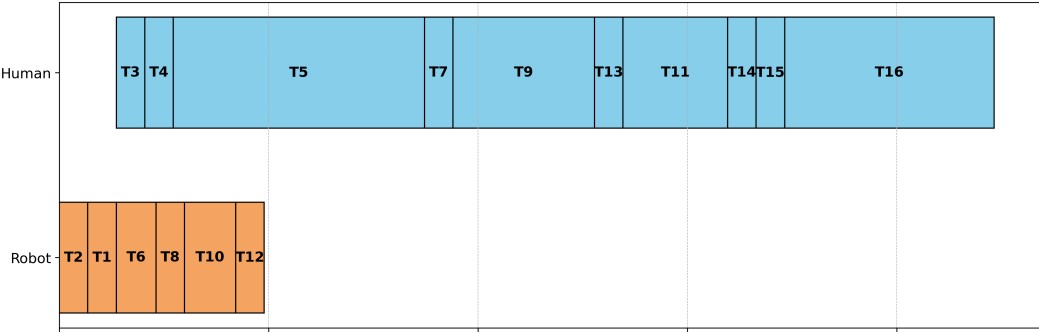

Figure 9: Initial schedule produced by HERON for cooking chicken with grilled vegetable (Task 3), showing parallel timelines for the human (blue) and robot (orange) agents. Sub-tasks are labeled **T1**–**T16** as follows: **T1** – pick up chicken, **T2** – pick up pot, **T3** – put chicken in pot, **T4** – put pot on stoveburner, **T5** – heat pot by stoveburner, **T6** – open drawer, **T7** – pick up knife, **T8** – pick up potato, **T9** – slice potato, **T10** – pick up tomato, **T11** – slice tomato, **T12** – pick up pan, **T13** – put potato in pan, **T14** – put tomato in pan, **T15** – put pan on stoveburner, **T16** – put pan on stoveburner.

**Experimental Log for Task 3 (Goal Change)**

[SimManager @ t=19.49s] Event: 'human_interrupt_goal_change' on Task: 'put_Chicken_in_Pot'
>>> INTERRUPT: Human requested a goal change: 'to cook chicken with grilled potato only'
— Triggering Dynamic Re-planning (Count: 1) —
    - Performing re-planning due to goal change.
    - New high-level goal: 'to cook chicken with grilled potato only'

[Optimizer] Solution found (Status: OPTIMAL).
— Optimal Schedule (from t=19.49s, Makespan: 1569.59s) —
    - open_Drawer | robot | 19.49 → 28.99
    - pickup_Knife | human | 28.99 → 35.79
    - pickup_Potato | robot | 28.99 → 35.79
    - put_Pot_on_StoveBurner | robot | 35.79 → 42.59
    - slice_Potato | human | 35.79 → 69.59
    - pickup_Pan | robot | 62.79 → 69.59
    - Heat_Pot_by_StoveBurner | human | 69.59 → 969.59
    - put_Potato_in_Pan | robot | 69.59 → 76.39
    - put_Pan_on_StoveBurner | robot | 962.79 → 969.59
    - Heat_Pan_by_StoveBurner | human | 969.59 → 1569.59

— Human's Ground Truth Behavior —
    - Task: pickup_Knife | Planned: 6.80s | Actual (GT): 5.89s | Actual End: 34.87s
    - Task: slice_Potato | Planned: 33.80s | Actual (GT): 29.26s | Actual End: 65.05s
    - Task: Heat_Pot_by_StoveBurner | Planned: 900.00s | Actual (GT): 779.22s | Actual End: 848.81s
    - Task: Heat_Pan_by_StoveBurner | Planned: 600.00s | Actual (GT): 519.48s | Actual End: 1489.07s

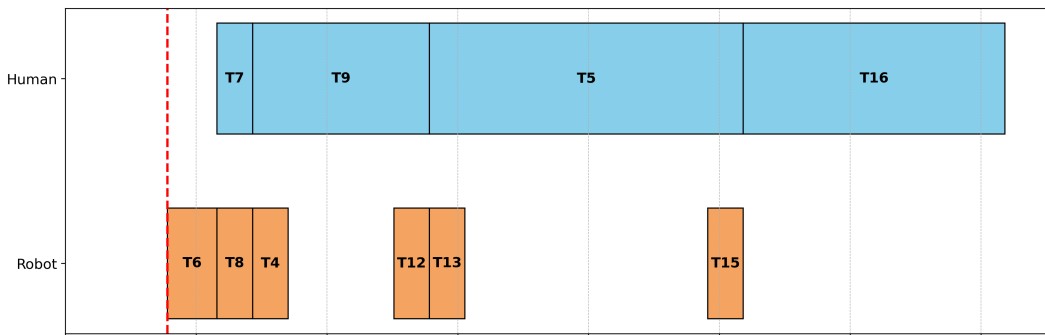

Figure 10: Updated schedule after the first re-planning caused by goal change. The human requested to cook chicken with grilled potato only and the schedule was recomputed to incorporate this new goal while preserving temporal consistency. Task labels follow the definitions given in Figure 9.

Experimental Log for Task 3 (Performance Variability)

[SimManager @ t=28.99s] Event: 'task_planned_end (robot)' on Task: 'open_Drawer'
>>> INTERRUPT: Human performance deviation detected.
— Triggering Dynamic Re-planning (Count: 2) —
    - Performing partial re-planning for remaining tasks.

[Optimizer] Solution found (Status: OPTIMAL).
— Optimal Schedule (from t=28.99s, Makespan: 1367.27s) —
    - pickup_Knife | robot | 28.99 → 35.79
    - pickup_Potato | robot | 35.79 → 42.59
    - slice_Potato | human | 42.59 → 68.56
    - pickup_Pan | robot | 42.59 → 49.39
    - put_Pot_on_StoveBurner | robot | 54.96 → 61.76
    - Heat_Pot_by_StoveBurner | human | 68.56 → 847.79
    - put_Potato_in_Pan | robot | 68.56 → 75.36
    - put_Pan_on_StoveBurner | robot | 75.36 → 82.16
    - Heat_Pan_by_StoveBurner | human | 847.79 → 1367.27

— Human's Ground Truth Behavior —
    - Task: slice_Potato_1 | Planned: 25.97s | Actual (GT): 25.97s | Actual End: 68.56s
    - Task: Heat_Pot_by_StoveBurner | Planned: 779.22s | Actual (GT): 779.22s | Actual End: 847.79s
    - Task: Heat_Pan_by_StoveBurner | Planned: 519.48s | Actual (GT): 519.48s | Actual End: 1367.27s

==================================================
EXPERIMENT FINISHED
==================================================

— FINAL RESULTS —
{
    "success": true,
    "final_makespan": 1367.2685597685158,
    "replanning_count": 2,
    "transport_rate": 1.0,
    "balance": 0.444,
    "total_steps": 13
}

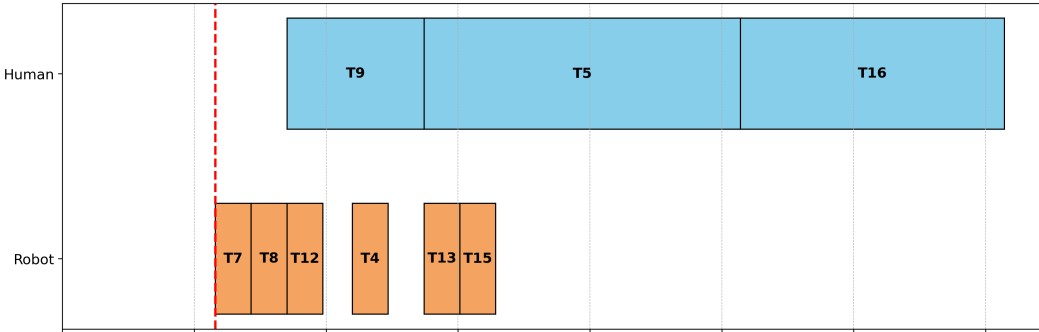

Figure 11: Updated schedule after the second re-planning triggered by performance deviation. Because the human completed the step faster than planned, the system classified the user as an expert and proportionally shortened the remaining plan times to accelerate overall execution. Task labels follow the definitions given in Figure 9.

## G.4 LOGS AND VISUALIZATIONS FOR TASK 4

**Experimental Log for Task 4**

```
==================================================
STARTING NEW EXPERIMENT
Goal: 'to make tomato pasta'
==================================================

— Initial Plan & Ground Truth —
— Optimal Schedule (from t=0.00s, Makespan: 643.00s) —
   - open_Drawer | robot | 0.00 → 3.00
   - pickup_Knife | human | 3.00 → 7.00
   - pickup_Tomato | robot | 3.00 → 8.00
   - slice_Tomato | human | 8.00 → 33.00
   - pickup_Bowl | robot | 8.00 → 13.00
   - pickup_Pasta | robot | 13.00 → 19.00
   - pickup_Pot | robot | 19.00 → 25.00
   - put_Tomato_in_Bowl | robot | 25.00 → 29.00
   - put_Pot_on_StoveBurner | robot | 29.00 → 34.00
   - put_Pasta_in_Pot | human | 33.00 → 37.00
   - Heat_Pot_by_StoveBurner | human | 37.00 → 39.00
   - Boil_Pasta_by_Pot | human | 39.00 → 639.00
   - put_Pasta_in_Bowl | robot | 639.00 → 643.00

— Human's Ground Truth Behavior —
   - Task: pickup_Knife | Planned: 4.00s | Actual (GT): 4.23s | Actual End: 7.23s
   - Task: slice_Tomato | Planned: 25.00s | Actual (GT): 26.42s | Actual End: 34.42s
   - Task: put_Pasta_in_Pot | Planned: 4.00s | Actual (GT): 4.23s | Actual End: 37.23s
   - Task: Heat_Pot_by_StoveBurner | Planned: 2.00s | Actual (GT): 11.24s | Actual End:
48.24s
   - Task: Boil_Pasta_by_Pot | Planned: 600.00s | Actual (GT): 633.99s | Actual End:
672.99s
```

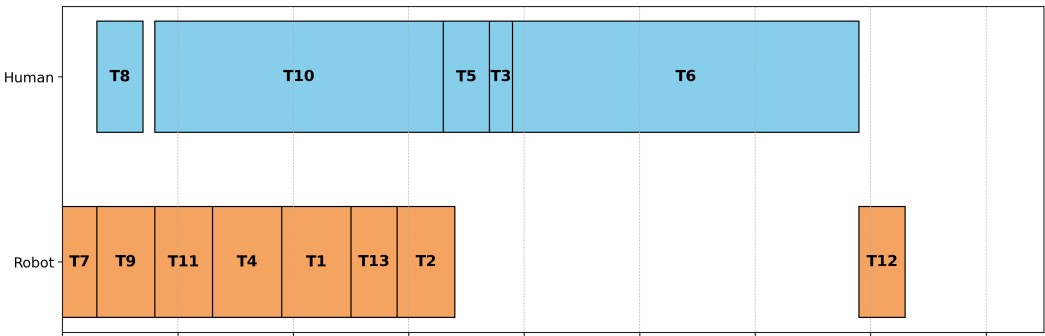

Figure 12: Initial schedule produced by HERON for making a tomato pasta (Task 4), showing parallel timelines for the human (blue) and robot (orange) agents. Sub-tasks are labeled **T1–T13** as follows: **T1** – pick up pot, **T2** – put pot on stoveburner, **T3** – heat pot by stoveburner, **T4** – pick up pasta, **T5** – put pasta in pot, **T6** – boil pasta by pot, **T7** – open drawer, **T8** – pick up knife, **T9** – pick up tomato, **T10** – slice tomato, **T11** – pick up bowl, **T12** – put pasta in bowl, **T13** – put tomato in bowl.

**Experimental Log for Task 4 (Performance Variability)**

[SimManager @ t=8.00s] Event: 'task_planned_end (robot)' on Task: 'pickup_Tomato'
>>> INTERRUPT: Human performance deviation detected.
— Triggering Dynamic Re-planning (Count: 1) —
    - Performing partial re-planning for remaining tasks.

[Optimizer] Solution found (Status: OPTIMAL).
— Optimal Schedule (from t=8.00s, Makespan: 677.52s) —
    - slice_Tomato | human | 8.00 → 34.42
    - pickup_Pasta | robot | 8.00 → 14.50
    - pickup_Pot | robot | 14.50 → 21.50
    - put_Pasta_in_Pot | robot | 21.50 → 28.00
    - put_Pot_on_StoveBurner | robot | 28.00 → 34.00
    - pickup_Bowl | robot | 34.00 → 40.00
    - Heat_Pot_by_StoveBurner | human | 34.42 → 36.53
    - Boil_Pasta_by_Pot | human | 36.53 → 670.52
    - put_Tomato_in_Bowl | robot | 40.00 → 46.00
    - put_Pasta_in_Bowl | robot | 670.52 → 677.52

— Human's Ground Truth Behavior —
    - Task: slice_Tomato | Planned: 26.42s | Actual (GT): 26.42s | Actual End: 34.42s
    - Task: Heat_Pot_by_StoveBurner | Planned: 2.11s | Actual (GT): 2.11s | Actual End: 36.53s
    - Task: Boil_Pasta_by_Pot | Planned: 633.99s | Actual (GT): 633.99s | Actual End: 670.52s

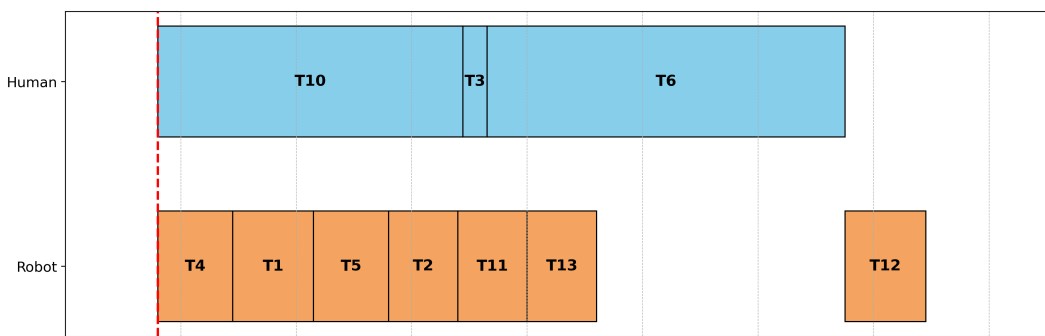

Figure 13: Updated schedule after the first re-planning triggered by performance deviation. Because the human's actual execution time exceeded the planned duration, the system classified the user as a novice and proportionally extended the remaining plan times to maintain overall feasibility. Task labels follow the definitions given in Figure 12.

**Experimental Log for Task 4 (Interruption)**

[SimManager @ t=38.03s] Event: 'human_interrupt_pause' on Task: 'Heat_Pot_by_StoveBurner'
>>> INTERRUPT: Human paused for 9.12s during task 'Heat_Pot_by_StoveBurner'.
— Triggering Dynamic Re-planning (Count: 2) —
    - Performing partial re-planning for remaining tasks.

[Optimizer] Solution found (Status: OPTIMAL).
— Optimal Schedule (from t=38.03s, Makespan: 683.65s) —
    - pickup_Bowl | robot | 38.03 → 40.00
    - put_Tomato_in_Bowl | robot | 40.00 → 44.00
    - Boil_Pasta_by_Pot | human | 47.15 → 679.65
    - put_Pasta_in_Bowl | robot | 679.65 → 683.65

— Human's Ground Truth Behavior —
    - Task: Boil_Pasta_by_Pot | Planned: 632.50s | Actual (GT): 632.50s | Actual End: 679.65s

==================================================
EXPERIMENT FINISHED
==================================================

— FINAL RESULTS —
{
    "success": true,
    "final_makespan": 683.646331069569,
    "replanning_count": 2,
    "transport_rate": 1.0,
    "balance": 0.444,
    "total_steps": 13
}

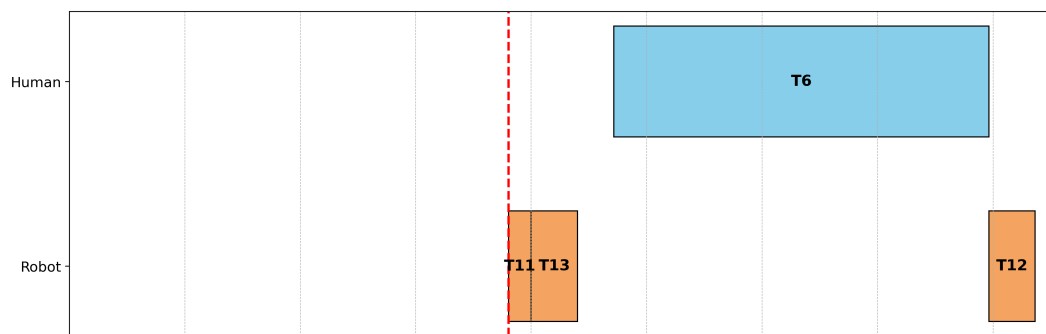

Figure 14: Updated schedule after the second re-planning caused by a temporary pause from human. Task labels follow the definitions given in Figure 12.

## H  DATASET-BASED EXECUTION TIME

Table 3: Dataset-based execution time for primitive sub-tasks used in our evaluation, representing averaged durations in seconds.

| Sub-task | Dataset | | | Average |
|---|---|---|---|---|
| | BRS | TeleMoMa | AIRoA MoMa | |
| pickup_<object_type> | 9.47 | 8.49 | 9.53 | 9.16 |
| put_<object_type>_on_<receptacle_type> | 8.89 | 9.77 | 11.54 | 10.07 |
| put_<object_type>_in_<receptacle_type> | 6.58 | - | 5.35 | 5.97 |
| open_<object_type> | 5.31 | 4.62 | 5.99 | 5.31 |
| close_<object_type> | 3.64 | - | 3.87 | 3.76 |

To support a quantitative evaluation of HERON's execution-time predictions, we compiled reference durations for primitive sub-tasks from publicly available mobile-manipulation datasets. These datasets include measured or human-calibrated execution times for common household actions such as picking, placing, opening and closing. For each action category that appears in our tasks, we extracted the corresponding durations and computed representative averages. The resulting values serve as dataset-based reference execution times for evaluating the generated schedules and provide an external point of comparison for the EET predictions used internally by HERON. These aggregated durations are summarized in Table 3.

- **BRS** (Jiang et al., 2025): The BEHAVIOR Robot Suite (BRS) provides a broad collection of execution data for household manipulation and navigation skills built on the BEHAVIOR benchmark. It contains measured and benchmarked durations for primitive actions such as picking, placing, reaching, grasping, opening, and closing, collected across various robot platforms in realistic indoor scenes. These timings serve as indicative references for the sub-task categories that appear in long-horizon mobile manipulation settings.
- **TeleMoMa** (Dass et al., 2024): TeleMoMa offers mobile manipulation demonstrations acquired through a flexible teleoperation framework that enables whole-body control on platforms including Tiago, HSR, and Fetch. The dataset reports action durations for manipulation and navigation primitives such as grasping, lifting, base adjustment, and arm–base coordination, obtained from both physical and simulated teleoperation runs. The resulting demonstrations provide detailed and human-calibrated timing profiles that align well with common sub-tasks in extended household workflows.
- **AIRoA** (Takanami et al., 2025): The AIRoA dataset presents large-scale real robot demonstrations of household mobile manipulation, supported by synchronized RGB video, proprioceptive feedback, and wrist force and torque sensing. It includes hierarchical annotations of short manipulation segments and basic action units, together with execution times for behaviors such as grabbing, opening, placing, and a variety of contact-driven movements. With more than ninety hours of demonstrations on the HSR platform, the dataset offers comprehensive and realistic timing statistics for evaluating sub-tasks in long household procedures.

## I  COMPUTATIONAL COST OF RE-PLANNING

Table 4: MILP re-planning time for each task and update count. Values outside the parentheses indicate the average solving time in seconds, and values inside the parentheses indicate the maximum solving time.

| Tasks | Number of Re-planning ($k$) | | |
|---|---|---|---|
| | 0 | 1 | 2 |
| Task 1 (Salad preparation) | 4.33 (5.47) | 3.71 (4.50) | 3.56 (4.13) |
| Task 2 (Baking) | 5.59 (6.94) | 5.29 (6.46) | 5.26 (6.24) |
| Task 3 (Chicken with grilled vegetables) | 5.11 (6.33) | 4.75 (5.71) | 4.40 (5.17) |
| Task 4 (Pasta preparation) | 4.67 (5.70) | 4.55 (5.41) | 4.05 (4.79) |

Table 4 reports the time required by the MILP scheduler to compute a schedule during both the initial planning step and the subsequent re-planning steps. For each of the four benchmark tasks, we report the average and maximum solver runtime across update counts $k \in \{0, 1, 2\}$, where $k = 0$ corresponds to the initial schedule and $k \geq 1$ corresponds to re-planning triggered by detected human uncertainty events. Across all tasks, the MILP solving time remains within a range from approximately 3.56 seconds to 5.59 seconds. All measurements were obtained using a machine equipped with an Intel Core i7-11800H CPU, and 16 GB RAM.

These measurements reflect the computational cost of scheduling task graphs that contain between 11 and 18 subtasks. Since the number of remaining subtasks decreases after each update, the runtime at later updates is generally slightly lower than the initial solve. The overall timescale of the re-planning process aligns with the decision points used by HERON. Re-planning is invoked only at the completion of subtasks or when the human collaborator reports an interruption or a change in goals, and these events typically unfold over several seconds or longer within long-horizon household tasks. The observed solver latency therefore matches the task-level re-planning granularity assumed by the framework.

Overall, Table 4 illustrates that, within the task sizes considered in our evaluation, the MILP scheduler exhibits stable and predictable runtime behavior. This enables HERON to perform dynamic re-planning without interrupting or delaying the natural progression of long-horizon human and robot collaboration.

## J FAILURE CASES

Despite the overall strong performance of HERON across diverse long-horizon kitchen tasks, several representative failures were observed that help reveal the framework's practical limitations. These cases highlight situations where the task-decomposition LLM, the physics-guided reasoning, or the scheduling optimizer interacted with human uncertainty in ways that reduced the success rate. Below we summarize the most salient failure modes identified during our experiments.

- **Invalid or Out-of-Scope Actions**: During experiments, the task decomposition LLM occasionally produced sub-tasks that fell outside the predefined HERON action space or referred to objects absent from the provided kitchen scenes. For instance, it sometimes suggested a *crack_egg* step even though "crack" was not among the allowed primitive actions, and in another case instructed the agent to locate and operate an oven that did not exist in the given scene metadata.
- **Over-Constrained Task Dependencies**: Among the long-horizon kitchen tasks with more sequential scenarios, the task decomposition LLM sometimes generated sub-tasks with unnecessarily strict or repetitive ordering constraints, forcing actions that could otherwise run in parallel to execute sequentially. Specifically, it required multiple prerequisite checks for the same step or linked independent sub-tasks into a single extended dependency path. These redundant constraints reduced opportunities for parallel execution and lengthened overall completion time, which in turn lowered the success rate.
- **Capability Misclassification in Physics-Guided LLM**: Despite explicit instructions in the physics-guided LLM to assign tasks according to each agent's physical limitations, in some cases the actions were not properly allocated to match those capabilities. For example, tasks that require fine dexterity or the use of sharp tools, such as slicing with a kitchen knife or operating a stove burner, were occasionally delegated to the robot even though these steps were explicitly intended for the human participant. This misallocation caused the resulting schedules to include steps the robot could not execute.
- **Uneven Agent Workload**: Although HERON is designed to minimize total completion time even at the cost of workload balance, some trials produced schedules in which all actions were assigned to the human while the robot remained consistently idle. From a pure optimization perspective this outcome still meets the goal of reducing the overall timespan, yet within the context of human–robot collaboration it should be regarded as a failure because the partnership becomes one-sided and the robot contributes little meaningful assistance.
- **Optimization Failure**: In a subset of trials, the MILP scheduler failed to return any feasible schedule even though the upstream task decomposition satisfied the required format. These failures stem from cases where the combined task constraints and time limits left no solution space for the optimizer, causing the execution to halt and directly reducing the success rate.

## K   VALIDATION SCOPE OF CONTRIBUTIONS

In this section, we provide a structured overview of which aspects of our stated contributions were fully validated in our experiments and which aspects remain only partially validated or left for future work. This is intended to clarify the scope of our claims and to support transparency in evaluation. Table5 summarizes the validation status.

Table 5: Overview of validated vs. not fully validated aspects of our contributions.

| Contribution | Validated in this paper | Not fully validated / Future work |
|---|---|---|
| **Dynamic Human-Uncertainty-Aware Scheduling** | • Implemented stochastic models of human uncertainty: performance variability ($\xi_1$), interruptions ($\xi_2$), and dynamic goal changes ($\xi_3$).
• Integrated uncertainty detection into the HERON pipeline, with dynamic re-scheduling demonstrated.
• Experiments in AI2-THOR kitchen scenarios show higher robustness and success rates compared to baselines (SMART-LLM, LiP-LLM, LLaMAR). | • Validation is limited to simulation; no human-in-the-loop user studies or physical robot experiments.
• Uncertainty models cover only three types (time variability, interruptions, goal changes), not more complex factors (e.g., fatigue, multitasking, evolving preferences).
• No quantitative analysis of how quickly or smoothly humans adapt to HERON's re-scheduling in practice. |
| **Physics-guided LLM for Feasible Task Allocation** | • Integrated robot kinematic and actuation limits into LLM-based reasoning.
• Showed that without physics-guided reasoning, allocations became unrealistic (e.g., assigning fine manipulation tasks to robots). | • Execution time estimates from the Physics-guided LLM were not quantitatively benchmarked against realistic measurements (e.g., real robot/human execution).
• Validation limited to a single domain (kitchen tasks in simulation). |
| **Hybrid LLM–Optimization Framework (LLM + MILP)** | • HERON consistently outperformed LLM-only baselines across four task types (parallel, sequential, hybrid).
• Ablation studies confirmed MILP's role: removing it increased execution timespan and reduced success rates.
• Demonstrated adaptability under uncertainty, showing improved robustness and efficiency. | • Validation conducted only in household kitchen scenarios; no experiments in other collaborative domains (e.g., manufacturing, logistics).
• Evaluations restricted to one human–one robot setup; scalability to multi-human or multi-robot teams remains unexplored.
• "Generalizability" claim is supported by multiple task structures but not by cross-domain empirical studies. |

Overall, the experiments validate that HERON:

• Achieves higher robustness under simulated human uncertainties,

• Produces more physically feasible and efficient task allocations through physics-guided reasoning,

• And benefits significantly from the integration of optimization (MILP) with LLM-based planning.

However, several aspects remain untested. Real-world human–robot experiments, systematic benchmarking of execution-time predictions, multi-agent scaling, and validation in domains beyond household tasks are important future directions to fully establish HERON's generality and practical utility.

## L    DECLARATION OF LLMS USAGE

LLMs were used in the preparation of this paper only as a general-purpose assist tool. Specifically, they were employed to polish the writing style, rephrase sentences for clarity, and check for consistency in notation and terminology. In addition, LLMs were used to help draft pseudocode and to convert author-written technical material into algorithmic or mathematical formulations; all such outputs were treated as editable suggestions and were thoroughly reviewed, verified against our implementations, and revised by the authors prior to inclusion. No part of the technical content, experimental design, or results was generated solely by LLMs. All ideas, formulations, experiments, and analyses are original contributions by the authors, who take full responsibility for the contents of the paper.

