# OpenReview forum: "HERON: Human-robot collaboration with Efficient and Resilient OptimizatioN for Long-horizon planning"
_ICLR.cc/2026/Conference — Submitted to ICLR 2026_

### Official Review · Reviewer_Qpzk · 2025-10-30

**Soundness:** 3
**Presentation:** 3
**Contribution:** 3
**Rating:** 6
**Confidence:** 4

**Summary:**

The paper introduces HERON which is a framework to create efficient and robust long-horizon schedules for human robot collaboration by explicitly accounting for human uncertainty. The framework consists of three sequential modules operating in an iterative loop:

1. Task Decompostion LLM: This translates natural language goal into a structured task graph of sub-tasks and dependencies.
2. Physics-guided LLM: Augments each sub-task with an estimated execution time and an optimal agent assignment, reasoning over physical constraints and cost.
3. MILP Optimizer: Generates an optimal schedule by solving a Mixed-Integer Linear Program that minimizes cost over makespan and workload distribution, while enforcing all temporal and resource constraints.

A continuous Verifying Stage monitors execution, detects human uncertainty events, and triggers dynamic re-planning by re-entering the planning loop. Experiments across four complex kitchen tasks in the AI2-THOR simulation demonstrate that HERON achieves a 45% higher success rate and a 13% reduction in schedule timespan compared to existing LLM-based planning baselines (SMART-LLM, LiP-LLM, LLaMAR)

**Strengths:**

1. The framework is fundamentally designed to be resilient and uncertainty-aware, explicitly modeling and dynamically re-scheduling in response to three categories of human stochasticity: performance variability ($\xi_1$), interruptions ($\xi_2$), and dynamic goal changes ($\xi_3$) which makes it more realistic.

2. The overall quality of the evaluation is strong, rigorously comparing HERON against multiple relevant LLM-based planning baselines (SMART-LLM, LiP-LLM, LLaMAR) and validating performance across four task structures (parallel, sequential, hybrid).

3. Ablation studies clearly demonstrate the necessity of both the Physics-guided LLM (although the validity of a physics-guided LLM in this paper is questionable, see weaknesses+questions) and the MILP Optimizer for sustaining high success rates and achieving temporal efficiency, confirming that dynamic planning alone is insufficient.

**Weaknesses:**

major weaknesses:

1. The paper acknowledges that the execution time estimates ($\hat{t}_i$) generated by the physics-guided LLM were not quantitatively benchmarked against realistic human or robot measurements. Without this quantitative validation, the claim that the LLM generates "realistic" times remains qualitative. Since the entire MILP optimization relies on these $\hat{t}_i$ values to minimize makespan, errors in the LLM's time estimates could lead the optimizer to generate a schedule that is optimal mathematically but suboptimal in reality.
2. The qualitative failure modes note that the Task Decomposition LLM sometimes produces unnecessarily strict or redundant dependency constraints. This forces parallelizable tasks to execute sequentially, which is likely the core reason why the total timespan (TI) for HERON in some complex tasks (like Task 3) is still quite long compared to the makespan of the purely symbolic LLM baselines (though the Success Rate is higher). This highlights an unaddressed brittleness in the initial LLM parsing step.

**Questions:**

1. Could the authors provide a simulation experiment where the $\hat{t}_i$ values generated by the Physics-guided LLM are replaced with values drawn from a Uniform distribution or a simpler analytical model (like the ones used in the prompts, $t=s/(2a)$ for robot travel), and compare the resulting metrics? This would quantitatively isolate the benefit provided by the LLM's "common-sense" time estimation from the pure benefit of the MILP solver.
2. How does the complexity of solving the MILP scale with the number of sub-tasks ($N$)? Since HERON performs dynamic re-planning based on perceived human behavior, the solver must return a new optimal schedule almost instantaneously. Could the authors provide re-planning time vs. $N$ plots to assure the real-time feasibility of the approach, or state the average and max computation times observed during dynamic re-planning?
3. The paper mentions that VLM-based monitoring to automatically detect performance Variability ($\xi_1$) lies beyond the current scope but is a future direction. Given that the human only provides feedback on actual completion time $t^{act}_i$ after the robot completes its first task, why not use a simpler form of VLM integration now? For instance, a VLM could visually monitor the human's progress on an assigned task (e.g., slicing a vegetable) to determine if they are currently engaged in the task, allowing for more proactive detection of a slow down $\xi_1$ *before* the robot completes its step.
4. While the $C(\mathcal{S})$ objective function includes a workload distribution term ($\lambda_{2} \cdot \sum \sum x_{i,a} \hat{t}_i(a)$), and the failure cases mention scenarios where the robot is idle, the resulting Balance (B) metric for HERON is often lower than LLaMAR's (e.g., Task 3: HERON is 0.56 vs. LLaMAR is 0.80). Please elaborate on the choice of weighting parameters $\lambda_1$ and $\lambda_2$. Were these chosen to strictly prioritize minimizing the makespan ($\lambda_1$) over achieving a balanced workload ($\lambda_2$), and would altering this trade-off significantly impact the overall success rate?

---

> ### Author Response · Authors · 2025-12-01
> **Response to Reviewer Qpzk**
>
> We thank the reviewer for the thorough and constructive evaluation of our work. We appreciate your recognition of HERON’s uncertainty-aware design, the strength of the evaluation across diverse task structures, and the insights provided by our ablation studies. We address all concerns and questions below.
>
> (The changes within the revised manuscript were highlighted in blue color.)
>
> ---
> 1. Quantitative validation of execution time estimates
>
> We agree with the reviewer that the earlier version of the manuscript did not provide a quantitative comparison between the execution time estimates generated by the physics-guided LLM and realistic reference values. In the revised version, we incorporated dataset-based execution durations extracted from existing mobile-manipulation datasets[1-3] and summarized them in **[Appendix H. Dataset-based Execution Time]**. These reference durations allow us to contextualize the scale of HERON’s predicted EET values and provide a grounded basis for evaluating scheduling performance. HERON continues to use its EET predictions internally for planning, while quantitative evaluation is performed using dataset-based time to ensure consistency and fairness across all frameworks. These durations were also used to update the Timespan (TI) values in Table 1 from **[6. Results and Discussion]**, which more clearly highlights HERON’s improvement in efficiency under long-horizon uncertainty.
>
> _Table 1. Dataset-based execution time (seconds) for primitive sub-tasks_
>
> | Sub-task                         | BRS  | TeleMoMa | AIRoA MoMa | Average |
> | -------------------------------- | ---: | -------: | ---------: | ------: |
> | `pickup_<object_type>`           | 9.47 | 8.49     | 9.53       | 9.16    |
> | `put_<object_type>_on_<receptacle_type>` | 8.89 | 9.77     | 11.54      | 10.07   |
> | `put_<object_type>_in_<receptacle_type>` | 6.58 | -        | 5.35       | 5.97    |
> | `open_<object_type>`             | 5.31 | 4.62     | 5.99       | 5.31    |
> | `close_<object_type>`            | 3.64 | -        | 3.87       | 3.76    |
>
> ---
> 2. Strict or redundant dependency constraints from the Task Decomposition LLM
>
> Thank you for highlighting this issue. The LLM-based task decomposition module can indeed introduce conservative or redundant dependency edges, particularly in long-horizon cooking scenarios that mix parallelizable and strictly ordered subtasks. These conservative relations may reduce the potential for parallelism. In our experiments, however, we observed that the efficiency improvements central to the proposed framework were largely preserved, as the MILP optimizer tended to reorganize executable portions of the task graph when feasible, thereby mitigating the practical impact of such conservative dependencies. We have clarified this interaction in the revised manuscript and explicitly acknowledged the brittleness identified by the reviewer. (**[4. Methodology – Task decomposition LLM]**)
>
> ---
> 3. Responses to the reviewer’s specific questions
>
> Q1. Replacing LLM-generated execution times with a simple analytical model
>
> Thank you for raising this point. We agree that replacing the LLM-generated execution times with a simple analytical model could serve as an ablation to evaluate the contribution of the physics-guided LLM. Several reviewers also noted that the original TI metric depended entirely on EET predictions, making efficiency comparisons largely relative rather than grounded in real task durations. In the earlier version of the paper, Timespan (TI) was computed directly from EET, which limited our ability to assess whether the predicted durations were physically meaningful beyond relative comparisons.
>
> To address this, we incorporated dataset-based execution times obtained from three widely used mobile-manipulation datasets and matched them to the subtasks appearing in our evaluation. These durations provide realistic ground-truth references, and we updated all TI values in the main results accordingly. Although the absolute TI values changed, HERON continued to produce the most efficient schedules across tasks, indicating that its improvements are not an artifact of EET-based scaling.
>
> This analysis demonstrates that the execution-time predictions produced by the physics-guided LLM are consistent with realistic task durations and that its contribution to HERON remains meaningful even when evaluated against grounded reference statistics.
>
> (Continued)

---

> ### Author Response · Authors · 2025-12-01
> **Response to Reviewer Qpzk (cont.)**
>
> Q2. Computational time for re-planning
>
> We have added average and maximum MILP solving times for all re-planning events in **[Appendix I. Computational Cost of Re-planning]**. Across update counts, solving time ranges from approximately 3.56 seconds to 5.59 seconds. Since re-planning is triggered only at sub-task boundaries or human-initiated events such as interruptions or goal changes, these decision points occur on a slower temporal scale. Sub-tasks in our experiments typically span tens of seconds or more, and entire tasks range from tens to over one thousand seconds. The solver latency is therefore compatible with the temporal granularity of HERON’s re-planning process. This distinction is now explicitly clarified in the revised manuscript.
>
> _Table 2. MILP re-planning time (seconds) for each task and update count.
> Values outside the parentheses indicate the average solving time, and values inside the parentheses indicate the maximum solving time._
>
> | Tasks                                   | 0 update            | 1 update           | 2 updates           |
> | --------------------------------------- | -----------: | -----------: | -----------: |
> | Task 1 (Salad preparation)              | 4.33 (5.47)  | 3.71 (4.50)  | 3.56 (4.13)  |
> | Task 2 (Baking)                          | 5.59 (6.94)  | 5.29 (6.46)  | 5.26 (6.24)  |
> | Task 3 (Chicken with grilled vegetables) | 5.11 (6.33)  | 4.75 (5.71)  | 4.40 (5.17)  |
> | Task 4 (Pasta preparation)               | 4.67 (5.70)  | 4.55 (5.41)  | 4.05 (4.79)  |
>
> Q3. Monitoring human’s utterance
>
> We agree that visual or multimodal monitoring could allow earlier detection of performance deviations or interruptions and reduce reliance on explicit human feedback. While this capability lies outside the scope of the current prototype, we outlined in the revised manuscript how VLM-based or multimodal perception modules could be incorporated in future extensions of HERON. (**[4. Methodology – Verifying Stage]**)
>
> Q4. About the weighting parameters for makespan and workload balance
>
> Thank you for pointing this out. In the early stages of system design, we considered incorporating a workload balance term and introduced the corresponding weighting parameters. Our initial intuition was that balanced task allocation might be desirable for collaborative settings. However, as we progressed, we realized that this assumption does not necessarily hold in human–robot collaboration, especially when the two agents differ substantially in capability or skill. In such cases, enforcing balance can actually conflict with the primary goal of achieving efficient and robust task execution.
>
> After further discussion among the authors, we decided to focus the framework on efficiency under uncertainty, which is the central contribution of HERON. As a result, the workload balance term and its weighting parameters were removed from the implementation. We appreciate the reviewer’s observation, which helped us identify that the earlier formulation could cause confusion. In the revised manuscript, we have removed the unused terms and streamlined the description of the scheduling objective to reflect the implementation actually used in our experiments. This correction ensures the presentation aligns clearly with the intended contribution of the work, which centers on improving efficiency in long-horizon human–robot collaboration.
>
> ---
> ### References
>
> [1] Yunfan Jiang, Ruohan Zhang, Josiah Wong, Chen Wang, Yanjie Ze, Hang Yin, Cem Gokmen, Shuran Song, Jiajun Wu, and Li Fei-Fei. Behavior robot suite: Streamlining real-world whole-body manipulation for everyday household activities. arXiv preprint arXiv:2503.05652, 2025.
>
> [2] Shivin Dass, Wensi Ai, Yuqian Jiang, Samik Singh, Jiaheng Hu, Ruohan Zhang, Peter Stone, Ben Abbatematteo, and Roberto Mart´ın-Mart´ın. Telemoma: A modular and versatile teleoperation system for mobile manipulation. arXiv preprint arXiv:2403.07869, 2024.
>
> [3] Ryosuke Takanami, Petr Khrapchenkov, Shu Morikuni, Jumpei Arima, Yuta Takaba, Shunsuke Maeda, Takuya Okubo, Genki Sano, Satoshi Sekioka, Aoi Kadoya, et al. Airoa moma dataset: A large-scale hierarchical dataset for mobile manipulation. arXiv preprint arXiv:2509.25032, 2025.

---

### Official Review · Reviewer_tXeB · 2025-11-01

**Soundness:** 3
**Presentation:** 2
**Contribution:** 2
**Rating:** 4
**Confidence:** 4

**Summary:**

This paper addresses the challenges of long-horizon planning in human-robot collaboration (HRC), specifically focusing on the inherent uncertainty of human partners, such as variable performance, unexpected interruptions, and dynamic goal changes. The authors propose HERON (Human-robot collaboration with Efficient and Resilient Optimization for Long-horizon planning), a novel framework that integrates LLMs with formal optimization techniques.

The HERON framework operates in a continuous loop: 1. Task Decomposition LLM first translates a human's natural language instruction into a set of structured sub-tasks with temporal dependencies. 2. Physics-guided LLM then enriches this plan by estimating execution times (EETs) and assigning each sub-task to either the human or robot, based on physical constraints, kinematics, and agent capabilities. 3. Mixed-Integer Linear Programming (MILP) Optimizer takes this enriched plan and generates a schedule that is optimized for efficiency.
4.  A Verifying Stage monitors execution. When it detects a human uncertainty event—either through verbal feedback from the human or by comparing EETs—it triggers a dynamic re-planning step, feeding updated information back to the optimizer to generate a new, resilient schedule.

Experiments conducted in the AI2-THOR simulator on complex, multi-step kitchen tasks demonstrate that HERON achieves higher success rates and more efficient scheduling (shorter timespans) compared to other LLM-based planning baselines.

**Strengths:**

A major strength is the framework's hybrid architecture. It intelligently leverages LLMs for their semantic understanding and decomposition capabilities (Stage 1) , while offloading the formal scheduling and constraint satisfaction to a classical MILP optimizer (Stage 3). This approach avoids the common pitfalls of pure-LLM planners, which often struggle with physical constraints, resource optimization, and efficiency . The ablation study (w/o-MILP) confirms that the optimizer is essential for achieving efficient schedules.

The concept of the "Physics-guided LLM" is a novel and powerful contribution. By forcing the LLM to reason about kinematics, actuation limits, and navigation costs before assigning tasks, the system generates plans that are physically feasible. The ablation (w/o-pLLM) strongly supports this, showing that a naive LLM assigns infeasible tasks to the robot (e.g., slicing, using fire), leading to a sharp decline in success rate.

The framework is not a static, one-shot planner. The verifying stage and re-planning loop make the system adaptive. The qualitative example in Figure 2, which shows the system gracefully handling both a performance deviation and an interruption   by re-scheduling the plan, is a compelling demonstration of the system's resilience .

**Weaknesses:**

The paper's most significant weakness is its evaluation. Despite being a paper on Human-Robot Collaboration, the framework was never tested with an actual human. The "human" is a stochastic simulation whose behavior is modeled by simple probability distributions (Normal, Bernoulli, Uniform) . The paper acknowledges this limitation in the appendix, stating, "Validation is limited to simulation: no human-in-the-loop user studies". This makes all claims about "resilience to human unpredictability" purely theoretical, as the simulated human is far less complex and more predictable than a real person.

The "Verifying Stage" is not an autonomous monitoring system. The paper's methodology reveals that detection of all three uncertainty types relies entirely on the human verbally communicating them to the robot.

The paper claims superior efficiency (Timespan) over baselines like SMART-LLM and LLaMAR. However, HERON is the only framework that incorporates a MILP optimizer specifically designed to "minimize makespan". The baselines are primarily LLM-driven planners that do not perform such rigorous optimization. The "w/o-MILP" ablation confirms this, showing that removing the optimizer significantly increases the Timespan. Therefore, the efficiency gains may not stem from a superior planning approach but simply from the fact that HERON uses an optimizer while the baselines do not.

**Questions:**

1. The current verifying stage seems entirely passive, requiring the human to verbally report all performance deviations, interruptions, and goal changes. How does this reliance on explicit human self-reporting scale to real-world scenarios where a human might be distracted, forget to report a delay, or not know how to quantify their own performance deviation?

2. This module estimates execution times by reasoning over physical constraints. Is the LLM performing symbolic calculations based on the provided physics formulas, or is it making "common-sense" estimations informed by those specs? If it's performing calculations, how do you ensure the LLM's numerical and physical reasoning is accurate and reliable?

---

> ### Author Response · Authors · 2025-12-01
> **Response to Reviewer tXeB**
>
> We sincerely thank the reviewer for the thoughtful and detailed evaluation. We appreciate your positive assessment of HERON’s hybrid architecture, the role of the physics-guided reasoning stage, and the adaptive behavior enabled by the re-planning loop. Below, we address each concern in detail.
>
> (The changes within the revised manuscript were highlighted in blue color.)
>
> ---
> 1. Lack of real human participants in the evaluation
>
> We agree that evaluation with real human users would provide stronger evidence for HERON’s resilience to human unpredictability. Practical constraints, including the availability of suitable robotic hardware, safety requirements, and the time needed for IRB approval, prevented such experiments within the project timeline. Our primary goal in this work was to validate the feasibility of HERON’s architecture in a controlled setting where human-related uncertainties can be specified, isolated, and reproduced. This environment allowed us to assess HERON’s efficiency and resilience under dynamically changing conditions induced by human variability.
>
> In the revised manuscript, we have clarified this limitation and outlined the requirements for future real-world studies. (**[7. Conclusion and Limitations]**)
>
> ---
> 2. Lack of quantitative evaluation of the EET module
>
> We acknowledge that relying on verbal reports is a simplifying assumption designed to isolate the scheduling and re-planning mechanisms from the complexities of autonomous monitoring. In the current prototype, explicit reporting ensures reliable detection of uncertainty events in simulation. However, more proactive and autonomous detection is feasible through multimodal perception, such as visual progress estimation or activity recognition.
>
> In the revised manuscript, we clarified that the verifying stage is not intended to represent the final monitoring solution and clarified how visual or multimodal modules could be integrated to automatically detect performance deviations or task engagement in real environments. (**[4. Methodology – Verifying Stage]**)
>
> ---
> 3. Efficiency comparison with baselines in the presence of the MILP optimizer
>
> We appreciate this important point. We agree that HERON’s improvement in Timespan (TI) is largely due to the presence of a dedicated makespan-oriented MILP optimizer, whereas the baselines rely solely on LLM-driven planning without explicit scheduling optimization. Our intention is not to claim that HERON is intrinsically more efficient than these baselines in a like-for-like comparison, but rather to highlight that combining LLM-based reasoning with a classical optimizer leads to more efficient execution plans in long-horizon settings with complex dependencies. HERON’s design follows a hybrid philosophy.
> - **LLM-based reasoning** is used to interpret natural-language instructions, derive hierarchical task structure, and incorporate physical and capability constraints through physics-guided reasoning.
> - **A classical MILP optimizer** then ensures that the resulting task graph is scheduled in a temporally and resource-consistent manner while minimizing the overall makespan and supporting dynamic re-planning.
>
>
> This hybrid design is central to our contribution, as our goal is to improve efficiency in long-horizon task planning by augmenting LLM-generated plans with optimization-based scheduling. Because existing baselines similarly use LLM-generated long-horizon plans, we consider them appropriate comparison points for illustrating the benefits of introducing a structured optimization layer.
>
> (Continued)

---

> ### Author Response · Authors · 2025-12-01
> **Response to Reviewer tXeB (cont.)**
>
> 4. Responses to the reviewer’s specific questions
>
> Q1. Scaling to real-world scenarios without explicit human reporting
>
> As mentioned above, we clarified that the current verifying stage uses explicit reporting to ensure reliable operation in simulation. Also, we indicated future extensions incorporating multimodal perception to enable more autonomous detection of human performance deviations. (**[4. Methodology – Verifying Stage]**)
>
> Q2. Whether the physics-guided LLM performs explicit numeric reasoning
>
> The physics-guided LLM does not perform explicit symbolic computation. Instead, it uses the provided physical parameters as a structural guide for reasoning about relative task difficulty, capability limits, and coarse execution time. To examine whether these predictions meaningfully reflect real task durations, we incorporated dataset-based execution times derived from three widely used mobile-manipulation datasets[1-3] and matched them to the primitive sub-tasks used in our evaluation. These dataset-based values offer realistic ground-truth references for contextualizing the scale of the predicted EET values.
>
> Although HERON continues to use its predicted EET internally for scheduling, Timespan (TI) is now quantitatively evaluated using the dataset-based execution times summarized in **[Appendix H. Dataset-based Execution Time]**, and all TI values in Table 1 from **[6. Results and Discussion]** have been updated accordingly. While this change affects the absolute TI values, the relative performance trends remain consistent: HERON continues to produce the most time-efficient schedules across all tasks. This outcome indicates that the execution-time estimates produced by the physics-guided LLM are physically reasonable in scale and that its contribution to the overall system remains meaningful even when compared against grounded reference statistics rather than solely against its own internal predictions.
>
> _Table 1. Dataset-based execution time (seconds) for primitive sub-tasks_
>
> | Sub-task                         | BRS  | TeleMoMa | AIRoA MoMa | Average |
> | -------------------------------- | ---: | -------: | ---------: | ------: |
> | `pickup_<object_type>`           | 9.47 | 8.49     | 9.53       | 9.16    |
> | `put_<object_type>_on_<receptacle_type>` | 8.89 | 9.77     | 11.54      | 10.07   |
> | `put_<object_type>_in_<receptacle_type>` | 6.58 | -        | 5.35       | 5.97    |
> | `open_<object_type>`             | 5.31 | 4.62     | 5.99       | 5.31    |
> | `close_<object_type>`            | 3.64 | -        | 3.87       | 3.76    |
>
> ---
> ### References
>
> [1] Yunfan Jiang, Ruohan Zhang, Josiah Wong, Chen Wang, Yanjie Ze, Hang Yin, Cem Gokmen, Shuran Song, Jiajun Wu, and Li Fei-Fei. Behavior robot suite: Streamlining real-world whole-body manipulation for everyday household activities. arXiv preprint arXiv:2503.05652, 2025.
>
> [2] Shivin Dass, Wensi Ai, Yuqian Jiang, Samik Singh, Jiaheng Hu, Ruohan Zhang, Peter Stone, Ben Abbatematteo, and Roberto Mart´ın-Mart´ın. Telemoma: A modular and versatile teleoperation system for mobile manipulation. arXiv preprint arXiv:2403.07869, 2024.
>
> [3] Ryosuke Takanami, Petr Khrapchenkov, Shu Morikuni, Jumpei Arima, Yuta Takaba, Shunsuke Maeda, Takuya Okubo, Genki Sano, Satoshi Sekioka, Aoi Kadoya, et al. Airoa moma dataset: A large-scale hierarchical dataset for mobile manipulation. arXiv preprint arXiv:2509.25032, 2025.

---

### Official Review · Reviewer_Gs3j · 2025-11-04

**Soundness:** 3
**Presentation:** 3
**Contribution:** 2
**Rating:** 6
**Confidence:** 4

**Summary:**

This paper presents HERON, a framework for long-horizon human–robot collaboration (HRC) that integrates large language models (LLMs), physics-based reasoning, and mixed-integer linear programming (MILP). HERON decomposes a natural language task into structured sub-tasks via an LLM-based task graph generator, estimates execution time and agent assignment using a physics-guided LLM, and computes an optimized task schedule with MILP. The system further monitors execution to handle human uncertainties such as performance variability, interruptions, and dynamic goal changes through re-planning. Experiments in simulated household tasks show improved task success rate and efficiency compared to existing LLM-based planners.

**Strengths:**

1. The paper formalizes LLM-based task decomposition into a structured graph representation, bridging natural language reasoning and symbolic planning.

2. The physics-guided time estimation and MILP optimization provide a principled way to achieve physically feasible and efficient task allocation between humans and robots.

3. The system’s ability to handle failures, interruptions, and goal changes through iterative re-planning is a meaningful advancement toward resilient HRC.

**Weaknesses:**

1. The task decomposition LLM relies on structured scene descriptions (object lists, coordinates) that must be provided explicitly. This limits HERON’s applicability to simulators or digital twins, since real-world visual perception rarely yields such clean symbolic input.

2. The time estimation (EET) module’s accuracy and reliability are not evaluated. There is no ablation or comparison between predicted and actual execution times, making it unclear how critical or accurate this module is to system performance.

3. The experiments are limited to simulation (AI2-THOR) with synthetic human models. The real-world robustness of HERON, especially given its reliance on structured metadata rather than sensory input, remains unproven.

**Questions:**

1. How robust is HERON if the environment description \$\mathcal{E}\$ is noisy or incomplete — for instance, if certain object attributes or coordinates are missing? Could the Task Decomposition LLM still produce usable task graphs?

2. Have you quantitatively evaluated the EET predictions against measured execution times in simulation?

3. How costly is each re-planning cycle?

4. Do you envision extending HERON to handle raw visual or multimodal inputs (e.g., VLM-based grounding) to overcome dependence on symbolic scene JSONs?

---

> ### Author Response · Authors · 2025-12-01
> **Response to Reviewer Gs3j**
>
> We sincerely thank the reviewer for the constructive and encouraging feedback. We appreciate your recognition of our efforts in formalizing LLM-based task decomposition, integrating physics-guided time estimation with MILP scheduling, and enabling resilient re-planning under uncertainty. We address the concerns below.
>
> (The changes within the revised manuscript were highlighted in blue color.)
>
> ---
> 1. Dependence on structured scene descriptions
>
> We agree that our current prototype assumes access to structured symbolic scene descriptions, including object identities and spatial attributes, to ensure reliable task decomposition. We have made this assumption explicit in the revised manuscript (**[4. Methodology – Task decomposition LLM]**). Our focus in this work is to evaluate HERON’s core contribution, which is a hybrid integration of LLM-based reasoning and classical optimization that improves efficiency and resilience in long-horizon collaboration. To isolate and rigorously assess the benefits of this hybrid architecture, we perform evaluations in a controlled simulation setting where scene information is clean and unambiguous. Introducing perception-related variability would obscure the evaluation of the reasoning and scheduling components that are central to our contribution, so we rely on clean and structured scene descriptions in this study.
>
> ---
> 2. Lack of quantitative evaluation of the EET module
>
> We appreciate the reviewer’s observation. In the revised manuscript, we address this point by incorporating dataset-based execution times derived from three widely used mobile-manipulation datasets[1-3] and matching them to the primitive sub-tasks that appear in our evaluation. These durations provide realistic ground-truth references for assessing how HERON’s EET predictions relate to actual task execution times. Although HERON continues to use its predicted EET internally for scheduling, Timespan (TI) is now quantitatively evaluated using dataset-based execution times, as summarized in **[Appendix H. Dataset-based Execution Time]**. We also updated the TI values reported in Table 1 from **[6. Results and Discussion]** to reflect this ground-truth-based model. While this change affects the absolute TI values, HERON continues to produce the most time-efficient schedules across all tasks, indicating that the execution-time predictions produced by the physics-guided LLM are consistent with realistic task durations and that its contribution to HERON remains meaningful even when evaluated against grounded reference statistics.
>
> _Table 1. Dataset-based execution time (seconds) for primitive sub-tasks_
>
> | Sub-task                         | BRS  | TeleMoMa | AIRoA MoMa | Average |
> | -------------------------------- | ---: | -------: | ---------: | ------: |
> | `pickup_<object_type>`           | 9.47 | 8.49     | 9.53       | 9.16    |
> | `put_<object_type>_on_<receptacle_type>` | 8.89 | 9.77     | 11.54      | 10.07   |
> | `put_<object_type>_in_<receptacle_type>` | 6.58 | -        | 5.35       | 5.97    |
> | `open_<object_type>`             | 5.31 | 4.62     | 5.99       | 5.31    |
> | `close_<object_type>`            | 3.64 | -        | 3.87       | 3.76    |
>
> ---
> 3. Simulation-only evaluation and absence of human participants
>
> We acknowledge this limitation. Conducting real-world experiments with human participants requires specialized robotic hardware, safety considerations, and IRB approval, which could not be arranged within the timeframe of this work. Our focus was to demonstrate the feasibility of HERON’s architecture, including its multi-stage reasoning pipeline, in enabling efficient scheduling and resilient handling of structured uncertainty. The revised manuscript clarifies this limitation and discusses the steps required for real-world deployment and user studies. (**[7. Conclusion and Limitations]**)
>
> (Continued)

---

> ### Author Response · Authors · 2025-12-01
> **Response to Reviewer Gs3j (cont.)**
>
> ---
> 4. Responses to the reviewer’s specific questions
>
> Q1. Robustness to noisy or incomplete scene descriptions
>
> As noted above, the current implementation operates under the assumption that the scene description is complete and structured. This design choice reflects the goal of evaluating the reasoning and scheduling pipeline independently of perception noise. We clarified this scope directly in (**[4. Methodology – Task decomposition LLM]**) of the revised manuscript. Handling noisy or incomplete perceptual information and coupling HERON with real-world grounding modules is a meaningful direction, but it falls beyond the scope of our present focus on validating the hybrid architecture itself.
>
> Q2. Quantitative evaluation of EET predictions
>
> As noted above, we added dataset-based execution durations as external references (**[Appendix H. Dataset-based Execution Time]**) and used these values to evaluate how HERON’s EET predictions align with realistic statistics.
>
> Q3. Cost of re-planning
>
> We now include detailed solver runtimes in **[Appendix I. Computational Cost of Re-planning]**. Across tasks and update counts, MILP solving time ranges from approximately 3.56 seconds to 5.59 seconds. Re-planning occurs only at sub-task boundaries or human-initiated events, which unfold on a slower timescale of tens of seconds or more. The solver latency is therefore compatible with the temporal granularity assumed by HERON’s decision process, and we clarified this point in the revised manuscript.
>
> _Table 2. MILP re-planning time (seconds) for each task and update count.
> Values outside the parentheses indicate the average solving time, and values inside the parentheses indicate the maximum solving time._
>
> | Tasks                                   | 0 update            | 1 update           | 2 updates           |
> | --------------------------------------- | -----------: | -----------: | -----------: |
> | Task 1 (Salad preparation)              | 4.33 (5.47)  | 3.71 (4.50)  | 3.56 (4.13)  |
> | Task 2 (Baking)                          | 5.59 (6.94)  | 5.29 (6.46)  | 5.26 (6.24)  |
> | Task 3 (Chicken with grilled vegetables) | 5.11 (6.33)  | 4.75 (5.71)  | 4.40 (5.17)  |
> | Task 4 (Pasta preparation)               | 4.67 (5.70)  | 4.55 (5.41)  | 4.05 (4.79)  |
>
> Q4. Extending HERON with multimodal perception
>
> We agree that this is a valuable direction. While beyond the scope of the current prototype, we made explicit how HERON could incorporate multimodal perception to increase autonomy and reduce reliance on symbolic metadata. (**[4. Methodology – Task decomposition LLM]**)
>
> ---
> ### References
>
> [1] Yunfan Jiang, Ruohan Zhang, Josiah Wong, Chen Wang, Yanjie Ze, Hang Yin, Cem Gokmen, Shuran Song, Jiajun Wu, and Li Fei-Fei. Behavior robot suite: Streamlining real-world whole-body manipulation for everyday household activities. arXiv preprint arXiv:2503.05652, 2025.
>
> [2] Shivin Dass, Wensi Ai, Yuqian Jiang, Samik Singh, Jiaheng Hu, Ruohan Zhang, Peter Stone, Ben Abbatematteo, and Roberto Mart´ın-Mart´ın. Telemoma: A modular and versatile teleoperation system for mobile manipulation. arXiv preprint arXiv:2403.07869, 2024.
>
> [3] Ryosuke Takanami, Petr Khrapchenkov, Shu Morikuni, Jumpei Arima, Yuta Takaba, Shunsuke Maeda, Takuya Okubo, Genki Sano, Satoshi Sekioka, Aoi Kadoya, et al. Airoa moma dataset: A large-scale hierarchical dataset for mobile manipulation. arXiv preprint arXiv:2509.25032, 2025.

---

### Official Review · Reviewer_KGwg · 2025-11-12

**Soundness:** 2
**Presentation:** 3
**Contribution:** 2
**Rating:** 2
**Confidence:** 5

**Summary:**

This paper presents a framework called HERON (Human-Robot Collaboration with Efficient and Resilient Optimization for Long-Horizon Planning) for long-horizon human-robot collaboration. The framework uses an LLM to decompose natural language task descriptions into subtasks and agent assignments, and another LLM to generate physics-guided execution time estimates and determine task assignments for the human and robot. The framework then uses a mixed integer linear programming (MILP) scheduler to dynamically reschedule task allocation based on observed human uncertainties in the form of variable performance, unexpected interruptions, and dynamic goal changes. The authors evaluate their framework on a set of human-robot collaboration settings by comparing performance with various LLM-based baselines and performing ablation studies.

**Strengths:**

(i) The paper focuses on the important problem of long-duration planning in human-robot collaboration, particularly in the context of multiple sources of uncertainty.

(ii) The framework considers various (kinematic, actuation, task dependency, feasibility) constraints in computing the initial ordering the tasks and an updated schedule when needed.

**Weaknesses:**

(i) The paper makes claims that are incorrect or unsubstantiated. For example, the paper identifies three sources of uncertainty associated with the human (variable performance, unexpected interruptions, and dynamic goal changes), but robots also exhibit these uncertainties. Also, there is a rich literature of prior work in robot planning that considers models of such uncertainties and replans as needed. Another example is the discussion of related work, where authors highlight the limitations of using LLMs to predict long-horizon plans but then do not consider the decades of research in robotics on generating such plans for robots collaborating with humans. In addition, the authors' approach to identifying uncertainty is essentially outcome monitoring and rescheduling, capabilities that are supported by many existing planners.

(ii) The justification for the design choices (made in this paper) is unclear; the approach seems to be to use LLMs for every possible component of the framework even when this is not really the correct choice. For example, the framework uses considerable prior knowledge of environment state and dynamics in the task decomposition LLM; if such prior knowledge exists, why use an LLM for the task decomposition instead of the many existing methods in classical planning that do not have the limitation displayed by LLMs in long-duration planning? The use of the physics-guided LLM makes even less sense; when the estimated execution time (EET) is based on kinematic and actuation constraints, why not use simple functions and probabilistic models of uncertainty instead of an LLM? In particular, the use of an LLM to compute values of parameters of various distributions is rather strange.

(iii) Following up on previous point, given the cost function used in this paper for the MILP scheduler, a classical/probabilistic AI planner could have been used instead of the framework described in this paper. Such a planner would have supported all the capabilities of the proposed framework while also providing performance guarantees and requiring much less resources (time, computation, storage etc).

(iv) The experimental evaluation is limited to LLM-based systems developed in the last couple of years; there is no attempt to compare performance with other ways in which the target problem could have been addressed.

**Questions:**

Please address questions and comments in the "Weaknesses" section above.

---

> ### Author Response · Authors · 2025-12-01
> **Response to Reviewer KGwg**
>
> We sincerely thank the reviewer for the detailed and thoughtful evaluation of our work. We appreciate the recognition that this paper addresses an important problem in long-horizon human–robot collaboration and that the proposed framework integrates constraints from kinematics, actuation, task dependency, and feasibility. We respond to the reviewer’s concerns below.
>
> (The changes within the revised manuscript were highlighted in blue color.)
>
> ---
> 1. Clarifying the sources of uncertainty in human–robot collaboration
>
> We agree that uncertainty may arise from both humans and robots. Our intention in this work is not to claim that robots lack uncertainty but to focus explicitly on human-driven variability. Human performance fluctuations, unexpected interruptions, and dynamic goal changes are qualitatively different from robotic execution noise and often more difficult to model using predefined symbolic descriptions. HERON aims to address these human-centered factors in long-horizon collaboration by integrating structured verification, physics-guided reasoning, and dynamic re-planning. This focus is already reflected in our problem formulation, and in the revised manuscript we made this scope more explicit by highlighting that HERON targets the specific challenges posed by human-centered uncertainty in long-horizon collaboration.
>
> ---
> 2. Positioning HERON with respect to classical planning and existing literature
>
> We appreciate the reviewer’s concern regarding the relationship between HERON and classical planning approaches. Classical symbolic planners and probabilistic AI planners provide strong foundations for reasoning about feasibility, constraints, and search. Our work does not intend to replace these methods. Instead, HERON occupies a complementary role by addressing challenges that arise in natural-language-defined, under-specified, and dynamically changing long-horizon tasks, which are difficult to specify within traditional operator-based planning domains.
>
> HERON’s design follows a hybrid philosophy.
> - **LLM-based reasoning** is used to interpret natural-language instructions, derive hierarchical task structure, and incorporate physical and capability constraints through physics-guided reasoning.
> - **A classical MILP optimizer** then ensures that the resulting task graph is scheduled in a temporally and resource-consistent manner while minimizing the overall makespan and supporting dynamic re-planning.
>
> This design allows HERON to combine the semantic and physical reasoning capabilities of LLMs with the formal optimization strength of classical schedulers, enabling effective operation in long-horizon HRC settings where symbolic domain models are not readily available.
>
> ---
> 3. Justification for using LLMs in task decomposition and time estimation
>
> We acknowledge the reviewer’s concerns regarding the use of LLMs. Our intention is not to replace established decomposition or temporal modeling techniques with LLMs where classical solutions are sufficient. Instead, LLMs are incorporated only in the components where their semantic reasoning ability provides advantages:
> - In **task decomposition**, the ability to infer hierarchical structures from natural language and context is essential because explicit operator models are not provided.
> - In **execution time estimation**, the physics-guided LLM is not used as a symbolic calculator, but as a reasoning scaffold that incorporates kinematic and actuation constraints to produce structured estimates. These predictions are now contextualized using dataset-based reference durations provided in **[Appendix H. Dataset-based Execution Time]**.
>
> Thus, LLMs serve as the semantic front-end of the system, extracting task structure and incorporating physical and capability constraints, while the classical MILP optimizer provides a schedule that respects resource constraints and temporal consistency, supporting efficient execution and dynamic re-planning.
>
> ---
> 4. Comparison with classical or probabilistic planners
>
> We agree that classical and probabilistic planners are powerful tools for long-horizon planning, and we appreciate the reviewer’s emphasis on this literature. However, direct comparison is not straightforward because these planners rely on complete symbolic domain models, deterministic or stochastic operator definitions, and well-structured preconditions and effects. In contrast, HERON tackles problems defined through natural-language instructions and evolving human behavior, where such domain specifications are not readily available.
>
> Therefore, our baselines consist of LLM-based long-horizon planners, which are the closest methodological comparators addressing similar task decomposition from high-level human instructions. At the same time, HERON incorporates a classical optimization module within the pipeline, reflecting a hybrid design that combines the semantic flexibility of LLMs with the structure and rigor of formal scheduling methods.
>
> (Contiuned)

---

> ### Author Response · Authors · 2025-12-01
> **Response to Reviewer KGwg (cont.)**
>
> 5. Scope of the experimental evaluation
>
> We acknowledge that real-world studies involving human participants would further strengthen the evaluation. Practical constraints related to hardware availability, safety requirements, and IRB approval prevented real-world experiments within the project timeline. Our primary goal in this work is to demonstrate the feasibility of HERON’s hybrid architecture and to evaluate its behavior under systematically controlled forms of human uncertainty.
>
> To address concerns about the depth of the evaluation, we have:
> - added dataset-based execution time references[1-3] to ground EET predictions **[Appendix H. Dataset-based Execution Time]**,
>
> _Table 1. Dataset-based execution time (seconds) for primitive sub-tasks_
>
> | Sub-task                         | BRS  | TeleMoMa | AIRoA MoMa | Average |
> | -------------------------------- | ---: | -------: | ---------: | ------: |
> | `pickup_<object_type>`           | 9.47 | 8.49     | 9.53       | 9.16    |
> | `put_<object_type>_on_<receptacle_type>` | 8.89 | 9.77     | 11.54      | 10.07   |
> | `put_<object_type>_in_<receptacle_type>` | 6.58 | -        | 5.35       | 5.97    |
> | `open_<object_type>`             | 5.31 | 4.62     | 5.99       | 5.31    |
> | `close_<object_type>`            | 3.64 | -        | 3.87       | 3.76    |
>
> - added detailed re-planning solver runtimes **[Appendix I. Computational Cost of Re-planning]**,
>
> _Table 2. MILP re-planning time (seconds) for each task and update count.
> Values outside the parentheses indicate the average solving time, and values inside the parentheses indicate the maximum solving time._
>
> | Tasks                                   | 0 update            | 1 update           | 2 updates           |
> | --------------------------------------- | -----------: | -----------: | -----------: |
> | Task 1 (Salad preparation)              | 4.33 (5.47)  | 3.71 (4.50)  | 3.56 (4.13)  |
> | Task 2 (Baking)                          | 5.59 (6.94)  | 5.29 (6.46)  | 5.26 (6.24)  |
> | Task 3 (Chicken with grilled vegetables) | 5.11 (6.33)  | 4.75 (5.71)  | 4.40 (5.17)  |
> | Task 4 (Pasta preparation)               | 4.67 (5.70)  | 4.55 (5.41)  | 4.05 (4.79)  |
>
> - clarified the effects of conservative dependencies in task decomposition (**[4. Methodology – Task decomposition LLM]**), and
> - strengthened discussions on future real-world extensions. (**[7. Conclusion and Limitations]**)
>
> ---
> 6. Closing
>
> We appreciate the reviewer’s feedback, which helped us refine the presentation of our contributions. HERON is built around a hybrid integration of LLM-based reasoning and classical optimization, enabling:
> - improved efficiency in long-horizon task execution, and
> - resilience to human-driven uncertainty through dynamic re-planning.
>
> The revised manuscript now more clearly articulates this hybrid perspective, the motivation for each component, and the boundaries of the study.
>
> ---
> ### References
>
> [1] Yunfan Jiang, Ruohan Zhang, Josiah Wong, Chen Wang, Yanjie Ze, Hang Yin, Cem Gokmen, Shuran Song, Jiajun Wu, and Li Fei-Fei. Behavior robot suite: Streamlining real-world whole-body manipulation for everyday household activities. arXiv preprint arXiv:2503.05652, 2025.
>
> [2] Shivin Dass, Wensi Ai, Yuqian Jiang, Samik Singh, Jiaheng Hu, Ruohan Zhang, Peter Stone, Ben Abbatematteo, and Roberto Mart´ın-Mart´ın. Telemoma: A modular and versatile teleoperation system for mobile manipulation. arXiv preprint arXiv:2403.07869, 2024.
>
> [3] Ryosuke Takanami, Petr Khrapchenkov, Shu Morikuni, Jumpei Arima, Yuta Takaba, Shunsuke Maeda, Takuya Okubo, Genki Sano, Satoshi Sekioka, Aoi Kadoya, et al. Airoa moma dataset: A large-scale hierarchical dataset for mobile manipulation. arXiv preprint arXiv:2509.25032, 2025.

---

### Meta-Review · Area_Chair_faZd · 2026-01-05

**Summary:**

This paper presents HERON, a framework for long-horizon human–robot collaboration (HRC) that integrates large language models (LLMs), physics-based reasoning, and mixed-integer linear programming (MILP). HERON decomposes a natural language task into structured sub-tasks via an LLM-based task graph generator, estimates execution time and agent assignment using a physics-guided LLM, and computes an optimized task schedule with MILP.  The authors evaluate their framework on a set of human-robot collaboration settings by comparing performance with various LLM-based baselines and performing ablation studies.

**Reviewer Concerns:**

The  Reviewer KGwg with score 2 is concerned about the incorrect or unsubstantiated claims, the unclear justification for the design choices, and the unclear motivation of why use an LLM for the task decomposition instead of the many existing methods in classical planning, and furthermore, the unclear advantage of the MILP scheduler compared with a classical/probabilistic AI planner.


The Reviewer tXeB with score 4 is mainly concerned about the evaluation: (i) the overly simplified "human" simulation modeled by simple probability distributions (Normal, Bernoulli, Uniform), (ii) The "Verifying Stage" is not an autonomous monitoring system. The paper's methodology reveals that detection of all three uncertainty types relies entirely on the human verbally communicating them to the robot.
(iii) the efficiency gains may not stem from a superior planning approach but simply from the fact that HERON uses an optimizer while the baselines do not.

The authors provide additional justification for these concerns, but some of them are not well addressed.

**Reviewer Scores:**

This is a borderline paper with scores 2, 4, 6, and 6.

After the rebuttal, the concerns about the evaluation and unclear advantage are still not well addressed.

---

### Decision · Program_Chairs · 2026-01-26

Reject